# An NKX2-1/ERK/WNT feedback loop modulates gastric identity and response to targeted therapy in lung adenocarcinoma

Rediet Zewdu[1,2], Elnaz Mirzaei Mehrabad[1,3], Kelley Ingram[1,4], Pengshu Fang[1,4], Katherine L Gillis[1,4], Soledad A Camolotto[1,2], Grace Orstad[1,4], Alex Jones[1,2], Michelle C Mendoza[1,4], Benjamin T Spike[1,4], Eric L Snyder[1,2,4]*

[1]Huntsman Cancer Institute, Salt Lake City, United States; [2]Department of Pathology, University of Utah, Salt Lake City, United States; [3]School of Computing, University of Utah, Salt Lake City, United States; [4]Department of Oncological Sciences, University of Utah, Salt Lake City, United States

**Abstract** Cancer cells undergo lineage switching during natural progression and in response to therapy. NKX2-1 loss in human and murine lung adenocarcinoma leads to invasive mucinous adenocarcinoma (IMA), a lung cancer subtype that exhibits gastric differentiation and harbors a distinct spectrum of driver oncogenes. In murine BRAF$^{V600E}$-driven lung adenocarcinoma, NKX2-1 is required for early tumorigenesis, but dispensable for established tumor growth. NKX2-1-deficient, BRAF$^{V600E}$-driven tumors resemble human IMA and exhibit a distinct response to BRAF/MEK inhibitors. Whereas BRAF/MEK inhibitors drive NKX2-1-positive tumor cells into quiescence, NKX2-1-negative cells fail to exit the cell cycle after the same therapy. BRAF/MEK inhibitors induce cell identity switching in NKX2-1-negative lung tumors within the gastric lineage, which is driven in part by WNT signaling and FoxA1/2. These data elucidate a complex, reciprocal relationship between lineage specifiers and oncogenic signaling pathways in the regulation of lung adenocarcinoma identity that is likely to impact lineage-specific therapeutic strategies.

*For correspondence:
eric.snyder@hci.utah.edu

Competing interests: The authors declare that no competing interests exist.

## Introduction

Lung adenocarcinoma (LUAD), the most common cause of cancer death worldwide, exhibits significant heterogeneity in tumor cell identity and overall differentiation state (*Travis et al., 2011*). The state of LUAD differentiation correlates closely with prognosis, intrinsic therapeutic sensitivity, and drug resistance (*Campos-Parra et al., 2014*; *Rotow and Bivona, 2017*; *Russell et al., 2011*). Recent work by our lab and others has shown that the pulmonary lineage specifier NKX2-1/TTF1 is a central regulator of LUAD growth and identity (*Maeda et al., 2012*; *Snyder et al., 2013*). NKX2-1 is expressed in ~75% of human LUAD, and NKX2-1 negative tumors confer a worse prognosis than NKX2-1-positive tumors (*Barletta et al., 2009*). The specific role of NKX2-1 in LUAD depends, in part, on the driver oncogene (*Maeda et al., 2012*; *Skoulidis and Heymach, 2019*; *Snyder et al., 2013*). The majority of LUADs harbor mutually exclusive mutations in driver oncogenes that signal through the mitogen-activated protein kinase (MAPK) pathway, including *EGFR* (14% of cases), *KRAS* (29%), and *BRAF* (7%) (early stage cases, reviewed in *Skoulidis and Heymach, 2019*). In this disease, active forms of the RAS family of small GTPases stimulate the RAF/MEK/ERK kinase cascade to drive proliferation, survival, and invasion.

Treatment-naive *EGFR*-mutant LUADs almost always express NKX2-1, and genetically engineered mouse models (GEMMs) suggest that NKX2-1 is required for optimal growth of *EGFR*-mutant LUAD

**eLife digest** When cells become cancerous they grow uncontrollably and spread into surrounding healthy tissue. As the cancer progresses different genes are switched on and off which can cause tumor cells to change their identity and transition into other types of cell. How closely tumor cells resemble the healthy tissue they came from can influence how well the cancer responds to treatment.

Many lung cancers have an identity similar to normal lung cells. However, some turn off a gene that codes for a protein called NKX2-1, which leads to a type of cancer called invasive mucinous adenocarcinoma (or IMA for short). Cells from this type of cancer develop an identity similar to mucous cells that line the surface of the stomach. But it was unclear how IMA tumor cells that developed from a mutation in the BRAF gene are affected by this loss in NKX2-1, and how transitioning to a different cell type impacts their response to treatment.

To investigate this, Zewdu et al. studied lung cells from patients with IMA tumors driven by a mutation in BRAF and cells from mice that have been genetically engineered to have a similar form of cancer. This revealed that the NKX2-1 protein is needed to initiate the formation of cancer cells but is not required for the growth of already established BRAF-driven tumors. Further experiments showed that removing the gene for NKX2-1 made these cancer cells less responsive to drugs known as BRAF/MEK inhibitors that are commonly used to treat cancer. These drugs caused the IMA cancer cells to change their identity and become another type of stomach cell. This identity change was found to depend on two signaling pathways which cells use to communicate.

This study provides some explanation of how IMA lung cancers that lack the gene for NKX2-1 resist treatment with BRAF/MEK inhibitors. It also shows new relationships between key genes in these cancers and systems for cell communication. These findings could lead to better therapies for lung cancer, particularly for patients whose tumor cells are deficient in NKX2-1 and therefore require specialized treatment.

(*Maeda et al., 2012*). Intriguingly, case reports suggest NKX2-1 expression can be lost when *EGFR*-mutant LUADs acquire drug resistance during a lineage switch to squamous cell carcinoma (*Levin et al., 2015*). In contrast to patient tumors that harbor *EGFR* mutations, loss of NKX2-1 expression is seen in a subset of *KRAS* and *BRAF* mutant tumors (*Skoulidis et al., 2015*; *Zhang et al., 2015*). In a *Kras^{G12D}*-mutant LUAD GEMM, *Nkx2-1* deletion enhances tumor growth and causes pulmonary to gastric transdifferentiation (*Snyder et al., 2013*), which is driven by the transcription factors FoxA1 and FoxA2 (*Camolotto et al., 2018*). Stochastic NKX2-1 loss has a similar effect on a distinct *Kras*-mutant LUAD GEMM (*Maeda et al., 2012*). *Nkx2-1*-deficient murine tumors closely resemble invasive mucinous adenocarcinoma (IMA), a subtype of NKX2-1-negative human LUAD that also expresses gastric markers. Human IMAs often harbor mutations in *KRAS* (62% of cases), occasionally *BRAF* (3%, including point mutations and fusions), but rarely *EGFR* (0.6%) (*Cha and Shim, 2017*). These studies suggest that there can be selective pressure for LUAD to either retain or downregulate NKX2-1 expression depending on the specific signaling networks activated by a given driver oncogene.

MAPK signaling is generally considered to drive proliferation and survival in LUAD, and has become a therapeutic target in some contexts (*Ferrara et al., 2020*). For example, small molecule inhibitors of BRAF and its downstream kinase MEK have recently shown clinical efficacy in *BRAF*-mutant LUAD (*Khunger et al., 2018*). However, mechanisms underlying heterogeneous responses remain to be identified. MAPK signaling also regulates differentiation state in a variety of tissues, including the stomach, depending on the magnitude and context of its activity (*Mendoza et al., 2011*; *Osaki and Gama, 2013*). We have previously shown that *Nkx2-1* deletion in KRAS^{G12D}-driven LUAD leads to IMA with high levels of MAPK activity, as assessed by levels of active (phosphory-lated) ERK1/2 (pERK). In contrast, *Nkx2-1* deletion in alveolar type-2 pneumocytes causes hyperplasia with low levels of pERK (*Snyder et al., 2013*). IMA and NKX2-1-negative hyperplasia both express some common pan-gastric markers (e.g. HNF4A), demonstrating that loss of NKX2-1 causes pulmonary to gastric transdifferentiation independent of elevated ERK activity. However, the cells in each lesion have a discrete morphology and form lesions with distinct architectures, suggesting that

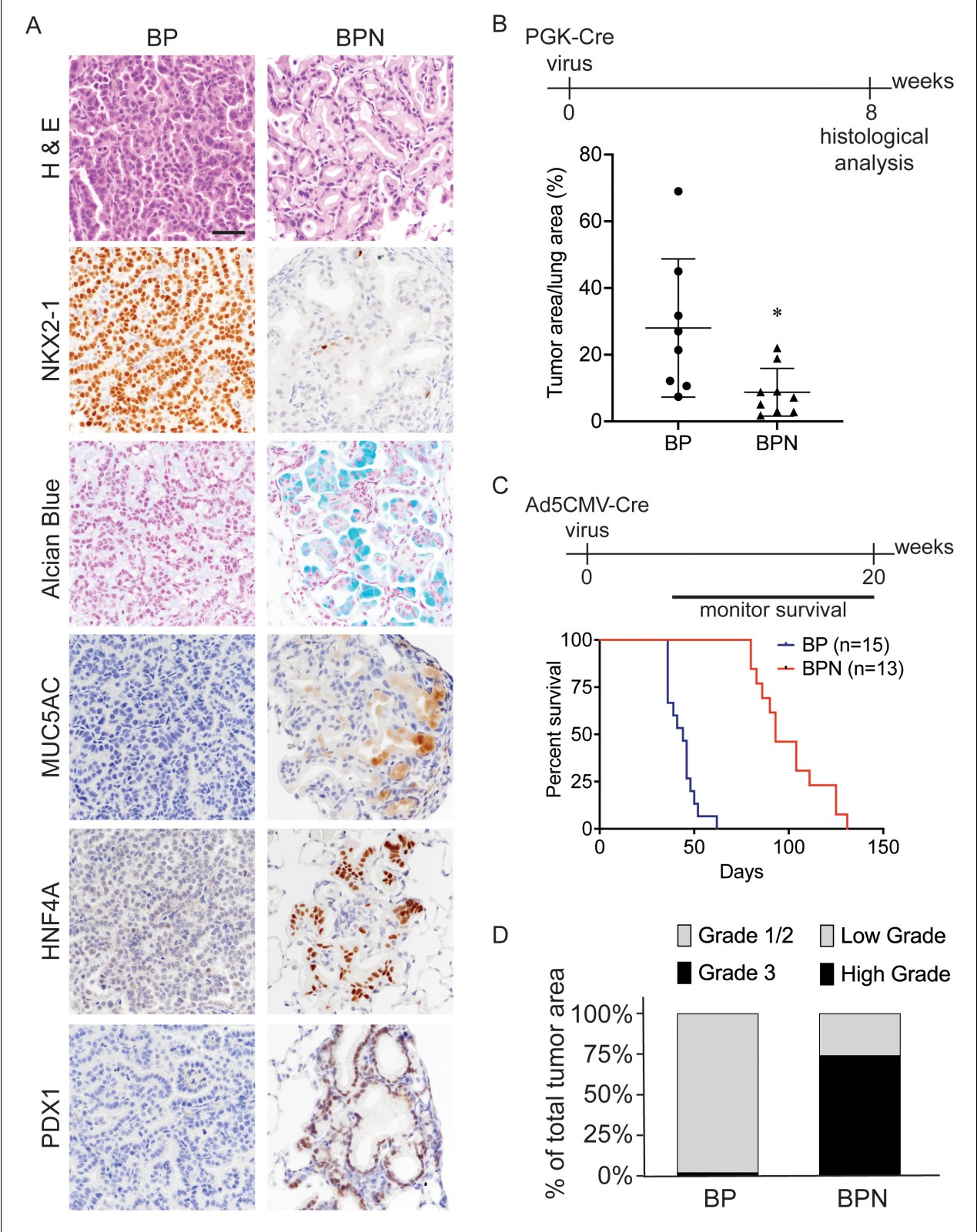

**Figure 1.** *Nkx2-1* deletion induces mucinous adenocarcinoma but impairs tumor initiation in BRAF$^{V600E}$-driven lung adenocarcinoma. (**A**) Representative photomicrographs of lung neoplasia arising 8 weeks after initiation with PGK-Cre lentivirus ($5 \times 10^3$ pfu/mouse). BP mice are *Braf$^{LSL-V600E/+}$;Trp53$^{f/f}$; Nkx2-1$^{f/+}$;Rosa26$^{LSL-tdTomato/LSL-tdTomato}$*. BPN mice are *Braf$^{LSL-V600E/+}$;Trp53$^{f/f}$;Nkx2-1$^{f/f}$;Rosa26$^{LSL-tdTomato/LSL-tdTomato}$*. Hematoxylin and eosin (H and E), Alcian Blue stain for mucin production, and immunohistochemistry (IHC) for NKX2-1, plus, markers of gastrointestinal differentiation. Scale bar: 100 μm. *Figure 1 continued on next page*

*Figure 1 continued*

(B) Quantitation of lung tumor burden 8 weeks after initiation with PGK-Cre lentivirus ($5 \times 10^3$ pfu/mouse) in indicated genotypes of mice: BP (n = 8), BPN (n = 9). *p=0.019 by Student's *t*-test. (C) Long-term survival after tumor initiation with Ad5-CMV-Cre adenovirus ($2.5 \times 10^7$ pfu/mouse) in mice of indicated genotypes. p<0.0001 by Log-rank test. (D) Histopathologic assessment of the percentage of tumors of indicated grade for mice enrolled in survival study (C).

The online version of this article includes the following figure supplement(s) for figure 1:

**Figure supplement 1.** *Nkx2-1* deletion induces mucinous adenocarcinoma but impairs tumor initiation in BRAF$^{V600E}$-driven lung adenocarcinoma.

**Figure supplement 2.** Percentage of incomplete recombinant tumors in *Braf*$^{LSL-V600E/+}$;*Trp53*$^{f/f}$;*Nkx2-1*$^{f/f}$;*Rosa26*$^{LSL-tdTomato/LSL-tdTomato}$ mice from *Figure 1C*.

oncogenic signaling downstream of KRAS$^{G12D}$ dictates the specific identity adopted by an NKX2-1-deficient lung cell.

We have sought to dissect the role of NKX2-1 in *Braf*-mutant LUAD using a GEMM of this disease. Here, we show that NKX2-1 regulates cellular identity, oncogenic signaling, and response to MAPK pathway inhibition in *Braf*-mutant adenocarcinoma. Our data show that the level of MAPK pathway activity dictates the specific identity adopted by NKX2-1-negative tumor cells within the gastric lineage. This identity shift is mediated in part by WNT signaling, which is increased after MAPK inhibition in IMA models.

## Results

### Activation of oncogenic BRAF in the absence of NKX2-1 from the pulmonary alveolar epithelium leads to the development of invasive mucinous adenocarcinoma

To dissect the role of NKX2-1 in mutant BRAF-induced lung adenocarcinoma, we utilized two established mouse strains bearing recombinase-activatable alleles of *Braf*$^{V600E}$. In the first mouse model, Cre-sensitive conditional alleles of *Braf*$^{LSL-V600E}$ (*Dankort et al., 2007*), *Trp53*$^{f/f}$ (*Jonkers et al., 2001*), *Rosa26*$^{LSL-tdTomato}$ (*Madisen et al., 2010*), and *Nkx2-1*$^{f/f}$ (*Kusakabe et al., 2006*) were recombined in the mouse lung epithelium upon intratracheal delivery of virus expressing Cre recombinase. Recombination of these alleles simultaneously activates BRAF$^{V600E}$ and tdTomato expression while eliminating p53 and NKX2-1 expression. Mice of the genotype *Braf*$^{LSL-V600E/+}$;*Trp53*$^{f/f}$;*Nkx2-1*$^{f/+}$; *Rosa26*$^{LSL-tdTomato/LSL-tdTomato}$ are hereafter referred to as BP mice and mice with the genotype *Braf*$^{LSL-V600E/+}$;*Trp53*$^{f/f}$;*Nkx2-1*$^{f/f}$;*Rosa26*$^{LSL-tdTomato/LSL-tdTomato}$ are hereafter referred to as BPN mice. Initial evaluation of histological (H and E) and molecular features revealed mucin production and gastrointestinal differentiation state in *Nkx2-1*-deficient tumors, including expression of HNF4A, PDX1, Gastrokine 1, Cathepsin E, and Galectin-4 (*Figure 1A*, *Figure 1—figure supplement 1A*). In lung tumors, NKX2-1 is required for sustained expression of pulmonary state marker genes such as those encoding the surfactant proteins (*Sftpb* and *Sftpc Snyder et al., 2013*) and the cell surface protein CD36 (*Camolotto et al., 2018*). Accordingly, immunohistochemical analysis showed that mucinous NKX2-1-negative tumors in BPN mice lack expression of pro-surfactant proteins B and C as well as CD36 (*Figure 1—figure supplement 1B*). Overall, NKX2-1-negative tumors in the *Braf*$^{LSL-V600E/+}$ model bore a close resemblance to those previously seen in the KRAS$^{G12D}$ GEMM and human IMA (*Snyder et al., 2013*).

Despite similar morphologic phenotypes in the BRAF$^{V600E}$ and KRAS$^{G12D}$ models, *Nkx2-1* deletion at tumor initiation had a profoundly distinct effect when combined with either of these oncogenes. Whereas *Nkx2-1* deletion augmented tumorigenesis initiated by KRAS$^{G12D}$ (*Snyder et al., 2013*), it significantly impaired the early stages of BRAFV600E-driven tumorigenesis. *Nkx2-1* deletion led to a lower tumor burden 8 weeks after initiation (*Figure 1B*, *Figure 1—figure supplement 1C*) and significantly prolonged overall survival (*Figure 1C*). We also quantitated tumor grade to better characterize tumor progression in this study and determine the predominant tumor type at the time of euthanasia. For BP tumors, we utilized the grading criteria described in *Winslow et al., 2011*. Almost all tumors were grade 1–2 (average ~98% of overall tumor burden), and grade three tumors were rare (*Figure 1D*). As BPN tumors are morphologically distinct from BP tumors, we needed to

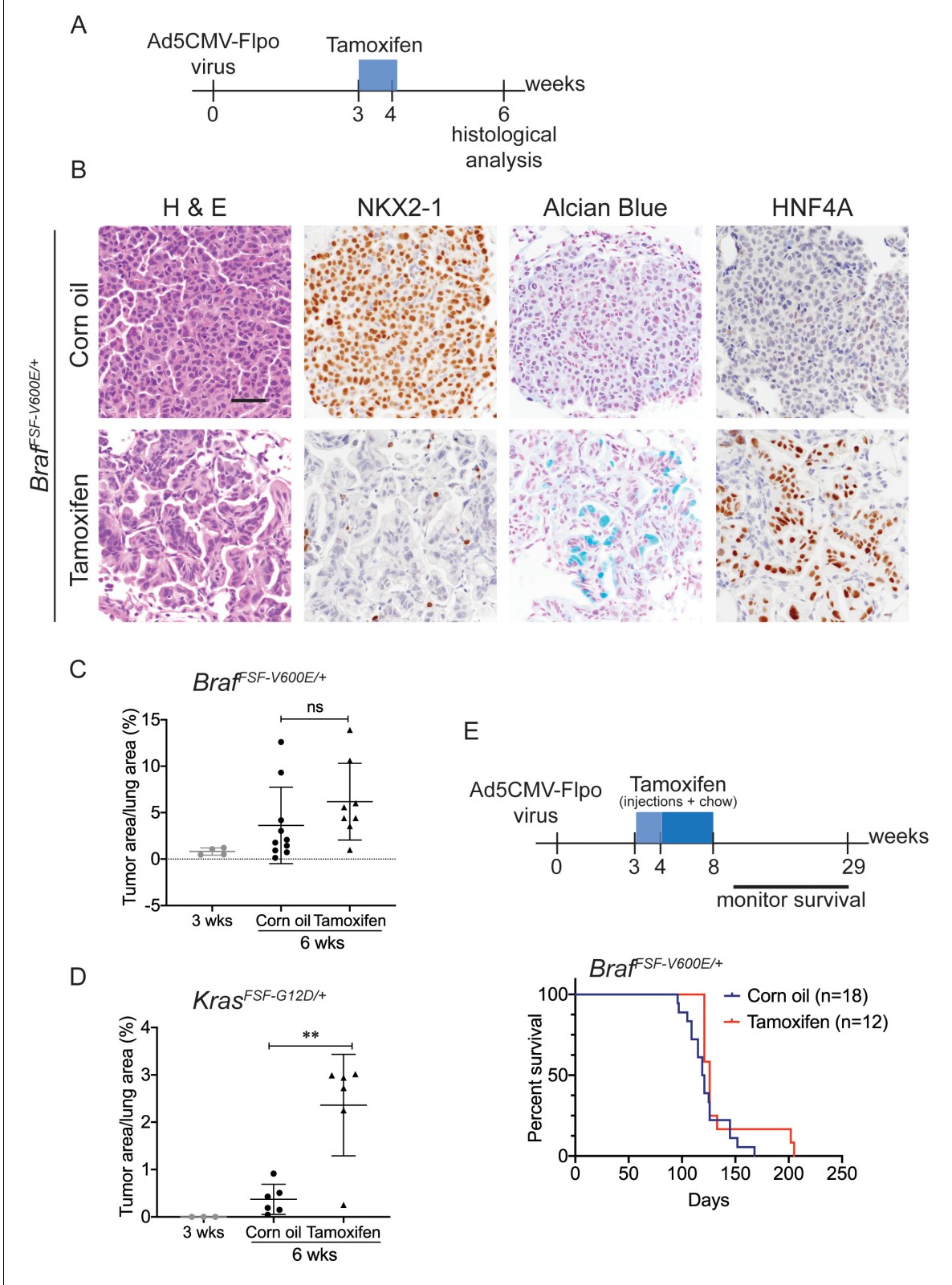

**Figure 2.** *Nkx2-1* deletion in established BRAF-mutant LUAD induces mucinous adenocarcinoma without altering tumor growth. (A–C) All mice are *Braf*^FSF-V600E/+^;*Trp53*^frt/frt^;*Nkx2-1*^f/f^;*Rosa26*^FSF-CreERT2/FSF-CreERT2^. Viral FlpO-recombinase expression simultaneously activated BRAF^V600E^ and Cre^ERT2^ and deleted *Trp53* by excision of *Frt* sites. Three weeks post-tumor initiation, mice were injected with corn oil (as vehicle control) or with tamoxifen to recombine *LoxP*-sites within *Nkx2-1* alleles by Cre^ERT2^ recombinase. Lungs were harvested 3 weeks thereafter. (A) Experimental scheme. (B) H and E,

*Figure 2 continued on next page*

*Figure 2 continued*

Alcian Blue, NKX2-1 and HNF4A staining of lung neoplasia arising 6 weeks after initiation with Ad5CMV-FlpO adenovirus ($2 \times 10^7$ pfu/mouse). Scale bar: 100 µm. (C) Quantitation of tumor burden at 3 weeks (n = 4 mice), and at 6 weeks after initiation, following corn oil treatment (n = 10 mice) or tamoxifen injections (n = 8 mice). Tamoxifen administration consisted of six intraperitoneal doses over 9 days. Graphs represent mean ± S.D. p Values are not statistically significant. (D) Mice are $Kras^{FSF-G12D/+}$;$Trp53^{frt/frt}$;$Nkx2-1^{f/f}$;$Rosa26^{FSF-CreERT2/FSF-CreERT2}$ and were administered with corn oil (n = 6 mice) or tamoxifen (n = 6 mice) injections as described in (A). Shown is the quantitation of tumor burden at 3 weeks and at 6 weeks post-initiation. Graphs represent mean ± S.D. **p=0.0014 by Student's $t$-test. (E) Survival of $Braf^{FSF-V600E/+}$;$Trp53^{frt/frt}$;$Nkx2-1^{f/f}$;$Rosa26^{FSF-CreERT2/FSF-CreERT2}$ mice that were treated with tamoxifen or vehicle starting at 3 weeks following tumor initiation. Tamoxifen administration consisted of six intraperitoneal doses over 9 days, followed by tamoxifen-containing chow for 1 month.

The online version of this article includes the following figure supplement(s) for figure 2:

**Figure supplement 1.** *Nkx2-1* deletion in established BRAF-mutant LUAD induces mucinous adenocarcinoma without altering tumor growth.

**Figure supplement 2.** Percentage of incomplete recombinant tumors in $Braf^{FSF-V600E/+}$;$Trp53^{frt/frt}$;$Nkx2-1^{f/f}$;$Rosa26^{FSF-CreERT2/FSF-CreERT2}$ mice that were treated with Tamoxifen from *Figure 2E*.

develop a grading scheme specifically for the cellular morphologies observed in this genotype. Accordingly, we characterize endpoint BPN tumors as 'low grade' if they are essentially identical to BPN tumors at early timepoints: tumor cells contain abundant intracellular mucin, low nuclear to cytoplasmic ratio, and small nuclei with minimal pleomorphism (*Figure 1A*). In contrast, we consider BPN tumors to be 'high grade' if they have diminished or absent mucin production, increased nucleus to cytoplasm ratio, and increased nuclear size and pleomorphism (*Figure 1—figure supplement 1D*). Based on these grading criteria, we found that high-grade BPN tumors comprised a much greater proportion of the NKX2-1-negative tumor burden than low-grade BPN tumors (~74% vs. 26%) (*Figure 1D*). Of note, a small minority of tumor burden (~4% on average) was NKX2-1-positive, indicative of a low rate of incomplete recombination (*Figure 1—figure supplement 2*). Thus, in this model, loss of NKX2-1 is compatible with tumor progression to a high-grade state, despite its impairment of tumor initiation. These data suggest that BP mice succumb primarily to an abundance of low-grade tumors. In contrast, the lower initial tumor burden in BPN mice provides more time for tumors to grow and progress to a high-grade state before compromising pulmonary function.

In the KRAS$^{G12D}$ model, we previously showed that inducing loss of lineage specifiers in established neoplasms can have distinct consequences when compared to lineage specifier loss at the time of tumor initiation (*Camolotto et al., 2018*). Models that enable gene manipulation in established tumors may be more physiologically relevant to tumor progression than models in which all genetic perturbations are present during tumor initiation. We therefore used a dual recombinase system in the present study to assess the role of NKX2-1 in the established BRAF$^{V600E}$ lung adenocarcinomas. Mice harboring FlpO-recombinase-sensitive alleles of $Braf^{FSF-V600E}$ (*Shai et al., 2015*), $Trp53^{frt/frt}$ (*Lee et al., 2012*), and $Rosa26^{FSF-CreERT2}$ (*Schönhuber et al., 2014*) were transduced with FlpO expressing adenovirus to recombine the above Flp-sensitive alleles and establish BRAF$^{V600E}$-induced lung lesions. After 3 or 6 weeks of tumor growth, $Nkx2-1^{f/f}$ alleles underwent Cre-based recombination upon injection of mice with tamoxifen (*Figure 2A*). To generate NKX2-1-positive controls, a cohort of mice with the same genotype were injected with vehicle (corn oil). In contrast to vehicle controls, *Nkx2-1* deletion drove transition of established tumors to invasive mucinous adenocarcinoma (IMA) (*Figure 2B*, *Figure 2—figure supplement 2A*). Collectively, these results and previous work from other labs indicate that *Nkx2-1* deletion promotes a cell lineage switch in LUAD driven by either KRAS$^{G12D}$ or BRAF$^{V600E}$ (*Maeda et al., 2012*; *Snyder et al., 2013*).

We next evaluated the effect of NKX2-1 loss on the growth of established BRAF$^{V600E}$-driven tumors. At early timepoints (3 weeks after tamoxifen treatment), *Nkx2-1* deletion had no significant effect on tumor burden (*Figure 2C*). In contrast, *Nkx2-1* deletion in established KRAS$^{G12D}$-expressing cells at the same time point greatly enhanced tumor burden, as described previously (*Figure 2D*; *Snyder et al., 2013*; *Young et al., 2011*). At this same timepoint, we also compared proliferation rates using MCM2 positivity in tumors and found that depletion of NKX2-1 did not impact cell proliferation (*Figure 2—figure supplement 1B,C*). Long-term survival analysis revealed no significant difference between tumor-bearing mice treated with tamoxifen and controls (*Figure 2E*). Histopathologic analysis of the survival study revealed that control mice harbored

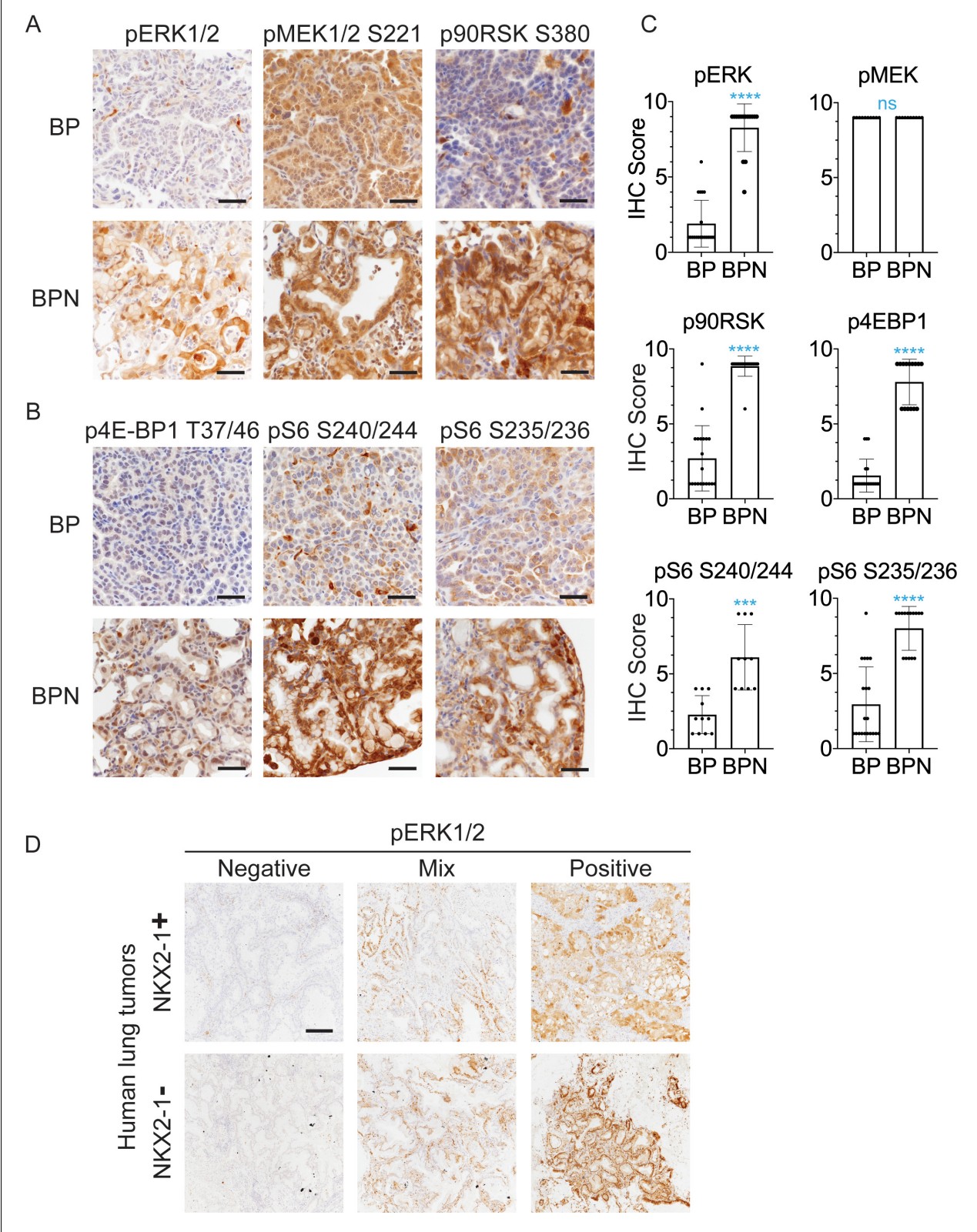

**Figure 3.** NKX2-1 regulates MAPK pathway in BRAF[V600E]-driven LUAD. (**A, B**) IHC for indicated proteins in lung neoplasia present at 8 weeks post-initiation in *Braf*[LSL-V600E/+]*;Trp53*[f/f]*;Nkx2-1*[f/+]*;Rosa26*[LSL-tdTomato/LSL-tdTomato] and *Braf*[LSL-V600E/+]*;Trp53*[f/f]*;Nkx2-1*[f/f]*;Rosa26*[LSL-tdTomato/LSL-tdTomato] mice. (**C**) IHC quantitation of phosphoproteins in indicated tumor types (n = 2–4 mice/genotype). Each dot represents one tumor. Error bars indicate mean ± S.

*Figure 3 continued on next page*

*Figure 3 continued*

D. ****p<0.0001, ***p<0.001, ns = not significant by Mann-Whitney test. (**D**) Representative staining of phospho-ERK in human non-mucinous (NKX2-1-positive, n = 51) and mucinous (NKX2-1-negative, n = 17) lung tumors. Scale bar: 500 µm.

The online version of this article includes the following figure supplement(s) for figure 3:

**Figure supplement 1.** NKX2-1 regulates MAPK pathway in BRAF^V600E^-driven LUAD.

predominantly non-mucinous NKX2-1-positive tumors, whereas tamoxifen-treated mice harbored predominantly NKX2-1-deficient tumors, many of which were high grade, further supporting the notion that NKX2-1 loss is permissive for malignant progression in BRAF-driven lung neoplasia (images and quantitation in *Figure 2—figure supplement 1D,E*). (A minority of tumor burden ~18% on average) was NKX2-1-positive in the survival study (*Figure 2—figure supplement 2*). These data show that loss of NKX2-1 in established BRAF$^{V600E}$ lung adenocarcinoma is tolerated but does not augment tumor growth, in contrast to NKX2-1 loss KRAS$^{G12D}$-driven tumors. This likely explains why *KRAS* mutations are enriched in IMA (relative to LUAD overall), whereas *BRAF* mutations are diminished in frequency, but not excluded altogether like *EGFR* mutations.

## Loss of NKX2-1 stimulates MAPK signaling downstream of mutant BRAF

We have previously shown that *Nkx2-1* deletion augments ERK activity in KRAS$^{G12D}$-driven lung adenocarcinoma (*Snyder et al., 2013*), and feedback inhibition of the MAPK pathway is known to be rate-limiting for the growth of KRAS$^{G12D}$-driven tumors in vivo (*Shaw et al., 2007*). We therefore asked whether the *Nkx2-1* deletion also alters MAPK activity in BRAF$^{V600E}$-driven lung tumors. We initially hypothesized that pERK levels would be similar in BP and BPN tumors, thus explaining the fact that *Nkx2-1* deletion does not augment BRAF$^{V600E}$-driven lung tumorigenesis. Surprisingly, *Nkx2-1* deletion in both the concomitant and sequential tumor models led to increased MAPK signaling downstream of oncogenic BRAF as assessed by IHC. Whereas NKX2-1-positive (BP) tumors generally exhibited weak, patchy staining for phosphorylated ERK1 and ERK2 (pERK), we observed strong pERK staining in NKX2-1-negative (BPN) tumors (*Figure 3A*, *Figure 3—figure supplement 1A*). In contrast, phosphorylated levels of MEK1/2 were similar in BP and BPN tumors, suggesting that activation of ERK1/2 in BPN tumors is largely due to regulation downstream of MEK1/2 activity (*Figure 3A*). Moreover, in BPN tumors, we found increased phosphorylation of p90RSK (a direct target of active ERK) and 4E-BP1 and S6 proteins - downstream targets of mTORC1, through which mTORC1 stimulates protein synthesis and cell growth (*Carriere et al., 2011*; *Mendoza et al., 2011*; *Roux et al., 2007*; *Figure 3A,B*; *Figure 3—figure supplement 1B*). We confirmed differences in degree of pathway activation between BP and BPN tumors by IHC quantitation (*Figure 3C*, *Figure 3—figure supplement 1B*). Therefore, BPN tumors exhibit hyperactivation of both MAPK and mTORC1 oncogenic pathways. Importantly, several of these distinguishing features observed in vivo, such as the association between low NKX2-1 and high pERK levels are conserved in vitro in primary tumor spheroid cultures from autochthonous BP and BPN tumors (*Figure 3—figure supplement 1C*).

Finally, we performed IHC for pERK on a panel of mucinous (n = 17, see Materials and methods for inclusion criteria) and NKX2-1-positive primary human lung adenocarcinomas (n = 51). Strong pERK staining was detectable in the majority of mucinous tumors (14/17). Among positive cases, seven tumors exhibited diffuse staining (>90% of tumor cells positive), and another seven exhibited partial positivity (20–90% of tumor cells). In contrast, NKX2-1-positive LUAD showed a trend toward lower pERK staining than the mucinous adenocarcinomas. Only 20% of NKX2-1-positive tumors were diffusely positive (vs. 41% of mucinous tumors), and 35% of NKX2-1-positive tumors were negative for pERK (vs. 18% of mucinous tumors). Quantitation and representative pictures are shown in *Figure 3D*, *Figure 3—figure supplement 1D*. The ERK phosphorylation site is more labile than epitopes recognized by antibodies that detect total protein levels. Therefore, staining patterns likely reflect both biologic heterogeneity and variable processing of clinical specimens. Nevertheless,

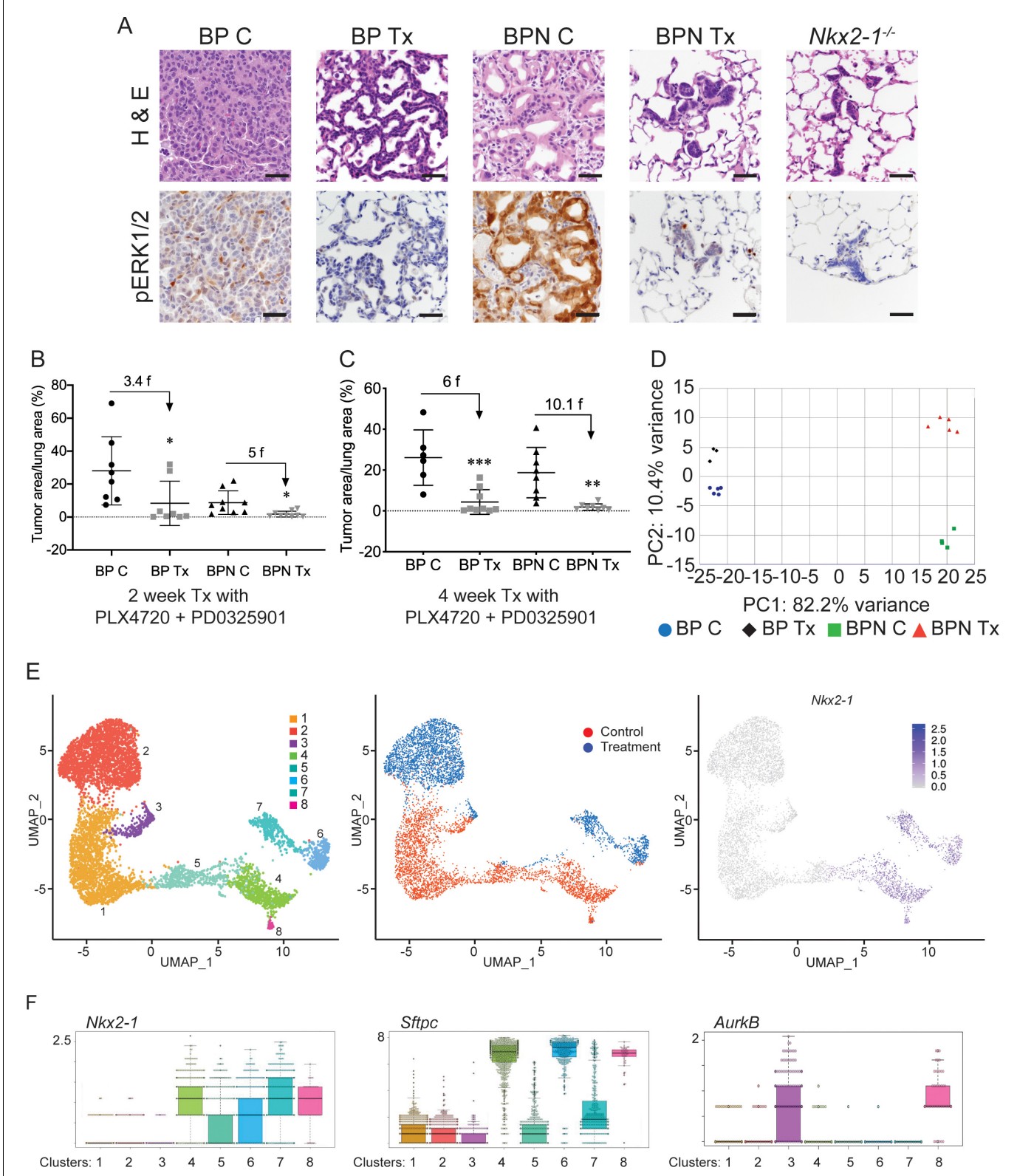

**Figure 4.** NKX2-1 status modulates response to MAPK pathway inhibitors. (A) Representative H and E and phospho-ERK1/2 immunostaining photomicrographs of paraffin-embedded lung sections from $Braf^{LSL-V600E/+};Trp53^{f/f};Nkx2-1^{f/+};Rosa26^{LSL-tdTomato/LSL-tdTomato}$ and $Braf^{LSL-V600E/+};Trp53^{f/f};Nkx2-1^{f/f};Rosa26^{LSL-tdTomato/LSL-tdTomato}$ mice that were treated with control chow (BP C/BPN C) or chow containing PLX4720 (200 mg/kg) and PD0325901 (7 mg/kg) inhibitors (BP Tx/BPN Tx) for 2 weeks starting at 6 weeks post-tumor initiation with PGK-Cre lentivirus ($5 \times 10^3$ pfu/mouse). Scale bar: 100 μm.
*Figure 4 continued on next page*

*Figure 4 continued*

(B, C) Quantitation of tumor burden in BP and BPN mice that were fed control chow or chow containing MAPK inhibitors starting at 6 weeks post-tumor initiation with PGK-Cre lentivirus ($5 \times 10^3$ pfu/mouse). Graphs represent mean ± S.D. (B) Chow treatment lasted 2 weeks and lungs were harvested at the 8-week timepoint. BP C (n = 8), BP Tx (n = 8), BPN C (n = 9), BPN Tx (n = 9). *p<0.05 by Student's *t*-test. Numbers indicated above graphs represent the fold reduction in tumor burden upon inhibitor-chow administration. (C) Chow treatment lasted 4 weeks and lungs were harvested at the 10-week timepoint. BP C (n = 6), BP Tx (n = 9), BPN C (n = 8), BPN Tx (n = 8). **p<0.01, ***p<0.001 by Student's *t*-test. Numbers indicated above graphs represent the fold reduction in tumor burden with inhibitor-chow administration. (D) Global gene expression analyses were performed on RNAs from FACS-sorted tdTomato$^+$ BP C (n = 5), BP Tx (n = 3), BPN C (n = 4), and BPN Tx (n = 5) murine lung tumor cells isolated at 7 weeks following initiation with Ad5-Spc-Cre adenovirus ($5 \times 10^8$ pfu/mouse for BP and $8 \times 10^8$ pfu/mouse for BPN mice). Control and MAPK-inhibitor chow treatments were given for 1 week at 6 weeks post-adenoviral instillation. Shown is the principal-component analysis (PCA) plot of the top 500 most variable genes showing that the four experimental groups of lung tumors, BP C, BP Tx, BPN C, and BPN Tx, had distinct global patterns of gene expression. (E) UMAP plots showing relatedness of high-quality, tumor cell scRNA-seq profiles from BPN control (n = 2) and BPN MAPKi-treated (n = 2) mice. Tumor cluster designations are indicated (left). Control and treated cells indicated (middle). *Nkx2-1* expression in scRNA-seq data indicating that clusters 4–8 represent incomplete recombinants (right). Single tumor cells were obtained by FACS-sorting tdTomato$^+$ cells isolated at 7 weeks following initiation with Ad5-Spc-Cre adenovirus ($8 \times 10^8$ pfu/mouse). Control and MAPK-inhibitor chow treatments were given for 1 week at 6 weeks post- adenoviral instillation. (F) Beeswarm plots of single cell sequencing data showing expression levels of *Nkx2-1*, *Sftpc* and *Aurkb* transcripts in tumor clusters 1–8. The online version of this article includes the following figure supplement(s) for figure 4:

**Figure supplement 1.** NKX2-1 status modulates response to MAPK pathway inhibitors.

these data show that the high levels of pERK in IMA mouse models can be observed in many cases of human IMA.

## Response of NKX2-1-positive and NKX2-1-negative tumors to RAF/MEK inhibition

The increase in MAPK activity we observed in both in vivo and in vitro models led us to ask whether NKX2-1 status influenced response to MAPK pathway inhibition. To investigate this, we directly compared the effect of RAF/MEK inhibitors on the growth and proliferation of autochthonous BP and BPN lung tumors. We chose dual inhibition of MEK and BRAF because this combination is the standard of care for BRAF$^{V600E}$-mutant human lung adenocarcinoma (*Planchard et al., 2016*) and because in other genetically engineered mouse models driven by BRAF$^{V600E}$ example for thyroid cancer (*McFadden et al., 2014*), the combination of the two inhibitors is more effective than either one alone at inhibiting the MAPK pathway in vivo. We found that administration of BRAF$^{V600E}$ inhibitor (PLX4720) in combination with MEK inhibitor (PD0325901) effectively suppressed ERK phosphorylation in lung tumors of both genotypes (*Figure 4A*). Combined BRAF/MEK inhibition led to a dramatically lower tumor burden in both BP and BPN mice when assessed after 2 weeks (*Figure 4B*) and 4 weeks (*Figure 4C*) of MAPKi treatment. The decrease in tumor burden in BPN mice was greater by two-fold compared to that in BP mice, suggesting that BPN tumors were overall more sensitive to MAPK inhibition. In fact, we noted that residual MAPK-inhibited BPN tumors resembled hyperplasia induced by *Nkx2-1* deletion alone with respect to both morphology and low pERK levels (*Figure 4A*). NKX2-1-positive, BRAF$^{V600E}$-driven lung tumors have been shown to regress when treated with MEK inhibitor, but regrow rapidly after drug cessation (*Trejo et al., 2012*). We therefore asked whether drug-treated residual BP and BPN cells retain tumorigenic potential. Histopathologic analysis of BP and BPN tumors treated for 4 weeks with BRAF/MEK inhibitor followed by drug removal showed that the residual cells readily grew back over the course of 1–4 weeks, adopting a morphology similar to untreated tumors (*Figure 4—figure supplement 1A*).

To gain insights into the mechanisms of drug response in each model, we performed transcriptome profiling by sequencing RNA from whole tumors and single-cell RNA sequencing. Using the Cre-activated *tdTomato* reporter (*Madisen et al., 2010*) in the *Braf*$^{LSL-V600E/+}$ model, we enriched tdTomato-positive tumor cells via fluorescence activated cell sorting (FACS). Tumor cells were isolated from mice of both genotypes (BP and BPN) that had been placed on BRAF/MEK inhibitor-diet or control chow for one week.

We first performed Principal Component Analysis (PCA) on RNA sequencing data from whole tumors using the top 500 most variable genes. This analysis revealed that replicate samples cluster together but that the four groups (control and MAPKi-treated BP and BPN tumors) could be

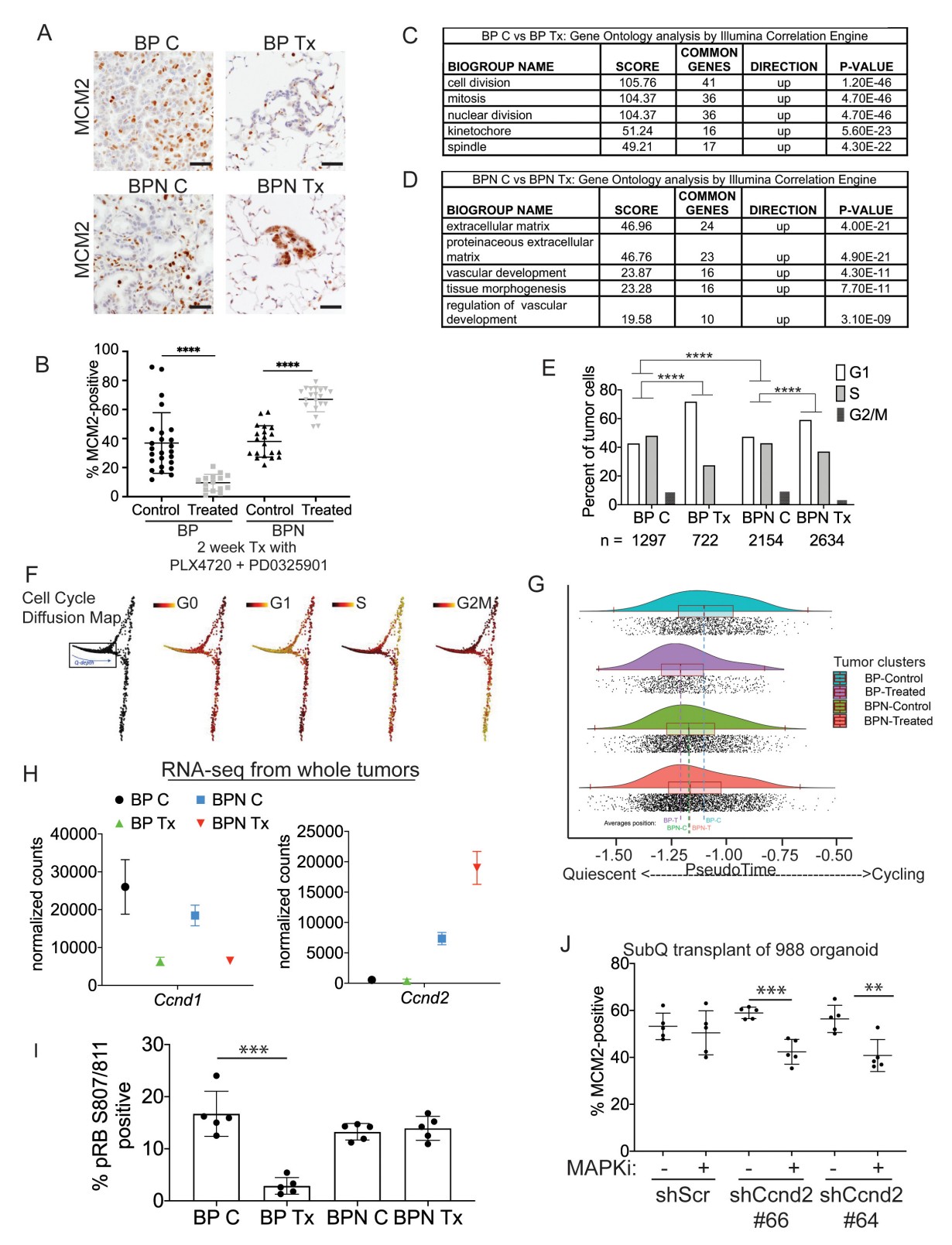

**Figure 5.** Differential impact of MAPK inhibitors on the cell cycle in NKX2-1-positive and NKX2-1-negative tumors. (A, B) Immunostaining for and quantitation of the proliferation marker MCM2 in tumors from BP and BPN mice at 8 weeks post-initiation. Control or MAPK-inhibitor infused chow feeding started at 6 weeks and was maintained for 2 weeks. BP C (25 tumors from four mice), BP Tx (15 tumors from three mice), BPN C (20 tumors from three mice), BPN Tx (20 tumors from three mice). Scale bar: 100 μm. Graphs represent mean ± S.D. ****p<0.0001 by Student's *t*-test. (C, D) Lists of
*Figure 5 continued on next page*

*Figure 5 continued*

the top-scoring Gene Ontology pathway terms of differentially expressed genes between control- and MAPKi drug-chow-treated BP (C) or BPN (D) tumors, as determined by Illumina Correlation Engine analyses of whole-tumor RNA-seq data. UP = enriched in control relative to treated samples. (E) Analysis of cell cycle score in single-cell RNA sequencing data using Seurat package. Utilized cell cycle genes as defined by *Mizuno et al., 2009*. ****p<0.0001 by Wilcox test. BP C = clusters 4, 5, 8; BP Tx = clusters 6, 7; BPN C = clusters 1, 3control; BPN Tx = clusters 2, 3treated (see *Figure 4E*). (F) Diffusion map of cell cycle phase signatures in scRNA-seq data showing coherent enrichment for phase specific signatures at specific graphical regions. > 80% of cells are disturbed in the box that correlates with G0 and G1 signatures and along which we fit a principle curve to model likely depth of quiescence (Q-depth). (G) Positioning of scRNA-seq profiles from the indicated cell clusters along the Q-depth curve. BPN control = cluster 1; BPN treated = cluster 2; BP control = cluster 4+5; BP treated = cluster 6+7. (H) Graph comparing absolute expression levels of *Ccnd1* and *Ccnd2*, data obtained from RNA sequencing of whole tumors. (I) Quantitation of phospho-RB-positive tumor cells in the indicated tumor types at 8 weeks post-initiation. Control or MAPK-inhibitor chow feeding started at 6 weeks and was maintained for 2 weeks. Graphs represent mean ± S.D. Multiple lesions per mouse, for five mice in each cohort, were analyzed. ***p<0.001 by Student's *t*-test. (J) Effect of knocking down CyclinD2 on MCM2 marker expression. A BPN organoid line was stably transduced with control or *Ccnd2*-targeting shRNA constructs followed by subcutaneous transplantation and MAPK-inhibitor chow treatment after 8 weeks of growth. Subcutaneous tumors were harvested after 1 month of drug treatment. MCM2 quantitation was confined to glandular structures that most closely resemble autochthonous BPN tumors. Graphs are mean ± S.D. ***p<0.001, **p<0.01 by Student's *t*-test. N = 5 mice per cohort.

The online version of this article includes the following figure supplement(s) for figure 5:

**Figure supplement 1.** Differential impact of MAPK inhibitors on the cell cycle in NKX2-1-positive and NKX2-1-negative tumors.

---

distinguished from each other at the transcriptomic level. As represented in *Figure 4D*, the greatest source of transcriptome diversity was related to deletion of *Nkx2-1* and is reflected by PC1. PC2 was related to MAPK inhibition-imposed transcriptomic changes, which appeared to more strongly distinguish BPN tumors than BP tumors. Regardless of treatment, BP tumors expressed higher levels of pulmonary genes (e.g. *Sftpb* and *Sftpc, Figure 4—figure supplement 1B*, *Supplementary file 1*) and BPN tumors expressed higher levels of gastric genes (e.g. *Pdx1* as well as a significant subset of HNF4A-target genes as revealed by the Illumina Correlation Engine [*Figure 4—figure supplement 1C*, *Supplementary file 1*]). Furthermore, *Dusp6*, a transcriptional readout of active ERK, was significantly downregulated in MAPK-inhibitor-treated BP and BPN tumors compared to their respective controls (*Figure 4—figure supplement 1D*, *Supplementary file 2*).

Despite these differences, NKX2-1 targets (e.g. *Sftpc*) were detected at higher levels than expected in sorted BPN tumor cells. We had previously observed that a minority of tumors in BPN mice were non-mucinous and retain NKX2-1 expression by IHC. We therefore inferred that a subset of these tumor cells may retain NKX2-1 due to incomplete recombination (*Figure 1* and data not shown). Since the presence of incomplete recombinants could compromise our ability to accurately analyze the transcriptional consequences of NKX2-1 deficiency, we performed additional single cell RNA-seq analysis of FACS-sorted BPN tumors (following 1 week of BRAF/MEK inhibitor or vehicle treatment) using the 10X Genomics platform. We predicted that this would enable us to circumvent the technical difficulty of resolving complete from incomplete recombinants and also investigate therapy response at the single-cell level.

We characterized the transcriptome of 5065 control and 5563 MAPKi drug-treated single BPN cells (*Figure 4—figure supplement 1E*). In our initial analysis, rare non-tumor cells clustered separately from tumor cells, and their identity was further validated by the expression of well-known stromal markers including *Vim*, *Ptprc*, *Trpm5*, *Pecam1*, *Mgp*, *Cd79a*, *Itgam*, *Adgre1*, *Cd3g*, and *Marco*. Filtering against contaminating stromal, endothelial, hematopoietic, and low-quality cells reduced the dataset to a total of 6807 high-quality tumor cells for further analysis (*Figure 4—figure supplement 1E*). We visualized these cells in reduced dimensionality using UMAPs and identified eight major tumor clusters (*Figure 4E*, *Supplementary file 3*). The majority of these cells (70%) fell into three NKX2-1-low clusters (1-3) and expressed multiple gastric markers (*Figure 4F*, *Supplementary file 3*, *4*). Examination of bam files confirmed that the rare *Nkx2-1* transcript counts in Clusters 1–3 correspond to residual flanking sequences from complete recombinants at the *Nkx2-1* locus and did not contain reads mapping to splice junctions of Cre-deleted exons in the *Nkx2-1* gene (*Figure 4—figure supplement 1F*). Cluster three exhibited high expression of *Aurkb*, *Plk1*, and other genes that suggest that these cells are in G2/M phase (*Figure 4F* and see *Figure 5* for

more detail). Cluster 1 and cluster 2 overwhelmingly contain cells from control and drug-treated mice, respectively.

In contrast, the remaining cells (30%) exhibited frequent and abundant *Nkx2-1* expression, including splicing from Cre-target exons indicating a failure to recombine (*Figure 4F*, *Figure 4—figure supplement 1F*). These cells also frequently expressed the NKX2-1 target *Sftpc* (*Figure 4F*). We interpret these cells, which fall into clusters 4–8, to be the incomplete recombinants that we have directly observed microscopically and inferred from RNA-seq data from whole tumors. The variable levels of *Nkx2-1* and *Sftpc* in clusters 4–8 (*Figure 4F*) suggests that some of these cells represent higher grade tumor cells that stochastically lose NKX2-1 activity and expression, as has been documented in KRAS^G12D-driven GEMMs (*Snyder et al., 2013*; *Winslow et al., 2011*). For example, we noted that cluster 5 cells express high levels of the embryonic marker *Hmga2*, a previously characterized marker of high-grade tumor cells (*Figure 4—figure supplement 1G*). Similar to cluster 3, cluster 8 exhibited high expression of *Aurkb*, *Plk1*, and other genes that suggest that these NKX2-1-positive tumor cells are in G2/M phase. In contrast to cluster 3, we note that cluster 8 is comprised almost exclusively of untreated cells, consistent with cell cycle exit of MAPKi-treated cells that retain NKX2-1 (*Figure 4E,F* and see *Figure 5* for further analysis). We quantified differences in MAPK pathway activation between BP and BPN tumors in our single-cell sequencing data by applying the MEK activity transcriptional signature published in *Dry et al., 2010*. This analysis substantiates our conclusions that BPN tumors display higher MAPK signaling than BP tumors (*Figure 4—figure supplement 1H*). Furthermore, using the Dry et al. signature, we find that RAF/MEK inhibition reduces pathway output in both genotypes (*Figure 4—figure supplement 1H*).

## NKX2-1 status influences cell cycle response to RAF/MEK inhibition

The MAPK pathway regulates proliferation by multiple mechanisms, including activating expression of the D-type cyclins, which can drive cells out of quiescence and into the cell division cycle (*Lavoie et al., 1996*; *Tuveson et al., 2004*). We therefore evaluated the impact of MAPKi drug treatment on cell cycle status of BP and BPN lung tumors using IHC for cell cycle markers as well as analysis of RNA-seq datasets. We first evaluated drug response by IHC for MCM2, a helicase detectable throughout the cell cycle but not in quiescence (G0). In BP tumors, 2 weeks of MAPKi treatment led to a significant decline in the percentage of MCM2-positive cells (*Figure 5A,B*). Additional IHC analysis showed a decline in the percentage of BP cells positive for BrdU incorporation and phospho-histone H3 (pHH3, M phase marker) (*Figure 5—figure supplement 1A–C*). Taken together, these data show that MAPKi treatment of BP tumors blocks proliferation and induces cell cycle exit, thereby increasing the proportion of cells in quiescence.

In contrast to BP tumors, MAPKi treatment of BPN tumors led to a paradoxical increase in the percentage of MCM2-positive cells (*Figure 5A,B*) despite decreased tumor burden (*Figure 4B*). In contrast, the percentage of BPN cells positive for BrdU and pHH3 declined to the same extent as BP cells after MAPKi treatment (*Figure 5—figure supplement 1A–C*). These data suggest that even though BPN cell proliferation is impaired by BRAF/MEK treatment, most residual BPN cells fail to exit the cell cycle and enter quiescence in the absence of MAPK activity.

Analysis of RNA-seq data from whole tumors further demonstrated that BP and BPN tumor cells exhibit a differential cell cycle response to MAPK inhibition. Drug-induced transcriptomic changes were highly distinct between genotypes. Pathway analysis (via Illumina Correlation Engine) demonstrated key differences between BP and BPN MAPK-inhibitor treated lung tumors in the most significant Gene ontology (GO) terms. Specifically, multiple cell cycle-related pathways decline in treated BP tumors, but not in treated BPN tumors, relative to their respective untreated controls (*Figure 5C*; *Supplementary file 5*, *6*). Instead, the top GO pathway terms in MAPK-inhibited BPN tumors relative to vehicle controls included pathways related extracellular reorganization and cell motility (*Figure 5D*; *Supplementary file 6*).

We calculated cell cycle scores of control and MAPKi-treated tumor cells in scRNA-seq data using the Seurat package and phased cell cycle gene signatures previously defined in mouse cells (*Mizuno et al., 2009*). The results of this analysis are consistent with the general conclusion that BP and BPN cells have distinct cell cycle responses to RAF/MEK inhibition (*Figure 5E*). However, the Seurat cell cycle score approach is not well adapted to distinguish G0 from G1 phase cells. We therefore developed a novel methodology for analyzing quiescence in scRNA-seq data using diffusion mapping (*Haghverdi et al., 2015*) and the complete set of fine-scale cell cycle phase signatures

from *Mizuno et al., 2009*, which includes both quiescent and cycling cells (*Figure 5—figure supplement 1D*). Using this novel approach, cells with high expression of S-phase or G2/M-phase signatures mapped to successive extremes at the right of the cyclic graph (*Figure 5F*), while the majority of cells (>80%) mapped to the left-hand portion corresponding to G0 and G1 signature enrichments (*Figure 5F*). Remarkably, these scRNA-seq analyses also revealed NKX2-1-dependent effects of drug treatment on cell cycle status. Treatment of BP cells caused a marked redistribution toward the left hand extreme of the graph, which is most distal from cycling cells and is enriched for the G0 signature, while the distribution of BPN cells was minimally affected by MAPK-inhibition (*Figure 5G*).

Given the ability of the MAPK pathway to drive cell cycle entry by activation of D-type cyclin expression, we evaluated the expression patterns of *Ccnd1-3* in whole tumor and single-cell RNA sequencing data. Suppression of MAPK activity led to decreased *Ccnd1* transcript and protein levels in both BP and BPN tumors (*Figure 5H*, *Figure 5—figure supplement 1E,F*) suggesting that the MAPK pathway is a major activator of *Ccnd1* expression in tumors of both genotypes. In contrast, *Ccnd2* mRNA levels were significantly higher in BPN tumors than BP tumors, and BRAF/MEK inhibition in BPN tumors led to a further increase in *Ccnd2* levels (*Figure 5H*, *Figure 5—figure supplement 1E*). At the protein level, we have only been able to detect Cyclin D2 by IHC in MAPKi-treated BPN tumors (*Figure 5—figure supplement 1G*). *Ccnd3* levels were low and relatively stable across all four conditions (*Supplementary file 1* and *4*).

These data raise the possibility that increased Cyclin D2 levels in BPN tumors upon MAPK inhibition might maintain CDK4/6 activity, thereby preventing cells from entering quiescence despite their MAPK-low state. To test this idea further, we used IHC to evaluate RB phosphorylation at S807/S811, two sites that can be phosphorylated by CDK4/6 (*Figure 5I*). Two weeks of combined BRAF/MEK inhibition significantly decreased the percent of phospho-RB-positive cells in BP tumors (*Figure 5I*). In contrast, MAPK-inhibition caused no change in phospho-RB S807/S811 levels in BPN tumors.

To evaluate specifically the potential role of Cyclin D2 in the cell cycle response to MAPK inhibitors, we knocked down Cyclin D2 in BPN organoids using two different shRNAs (*Figure 5—figure supplement 1H*). We then transplanted these lines into NSG mice subcutaneously, allowed tumors to form and grow for 8 weeks, and treated mice with RAF/MEK inhibitor for 4 weeks. Histopathologic analysis revealed that organoid-derived tumors exhibited a triphasic morphology in vivo, including large cystic structures, glandular adenocarcinoma, and small foci of poorly differentiated adenocarcinoma. Cystic structures appeared to accumulate fluid at a variable rate, precluding the use of tumor size as a metric of drug response. We quantitated MCM2 levels in the glandular adenocarcinomas, the component with the greatest similarity to the autochthonous model. We found that MAPK inhibition failed to reduce the MCM2 rate in control adenocarcinomas, whereas the MCM2 rate declined in tumors with Cyclin D2 knockdown upon RAF/MEK inhibitor treatment (*Figure 5J*). Although organoid-derived subcutaneous tumors do not perfectly mimic the autochthonous BPN model, these data provide additional orthogonal evidence that Cyclin D2 plays a role in the response of NKX2-1-negative lung adenocarcinoma to MAPK inhibition.

Taken together, these data show that NKX2-1 status has a profound effect on the cell cycle response of BRAF^V600E-driven lung tumors to targeted therapy. We propose that Cyclin D2 helps maintain CDK4/6 activity and RB phosphorylation in MAPKi-treated BPN tumors, thus preventing cell cycle exit. The inability of residual BPN cells to exit the cell cycle suggests that distinct therapeutic strategies may be needed to eliminate BP versus BPN tumor cells after MAPK inhibition.

## MAPK pathway regulates identity of NKX2-1-negative mucinous adenocarcinoma within the gastric lineage

Further analysis of gene expression changes in BPN tumors revealed surprising switch-like changes in the expression of lineage markers associated with specific gastric cell types. BPN tumor cells normally express high levels of transcripts that mark surface mucous cells in the stomach, such as *Muc5ac* and *Gkn1*. We find that these surface mucous cell markers decline significantly upon MAPK pathway inhibition (*Figure 6A*). In parallel, two of the most highly upregulated genes in RAF/MEK inhibitor-treated BPN tumor cells were *Pgc* and *Cym* (*Figure 6A*, *Figure 4—figure supplement 1D*), which encode digestive enzymes normally expressed in chief cells found at the base of the stomach gland (*Han et al., 2019*; *Leushacke et al., 2017*; *McCracken et al., 2017*). Additional markers of murine gastric chief cells including *Pga5* and *Clps* were also induced by MAPK inhibition in these

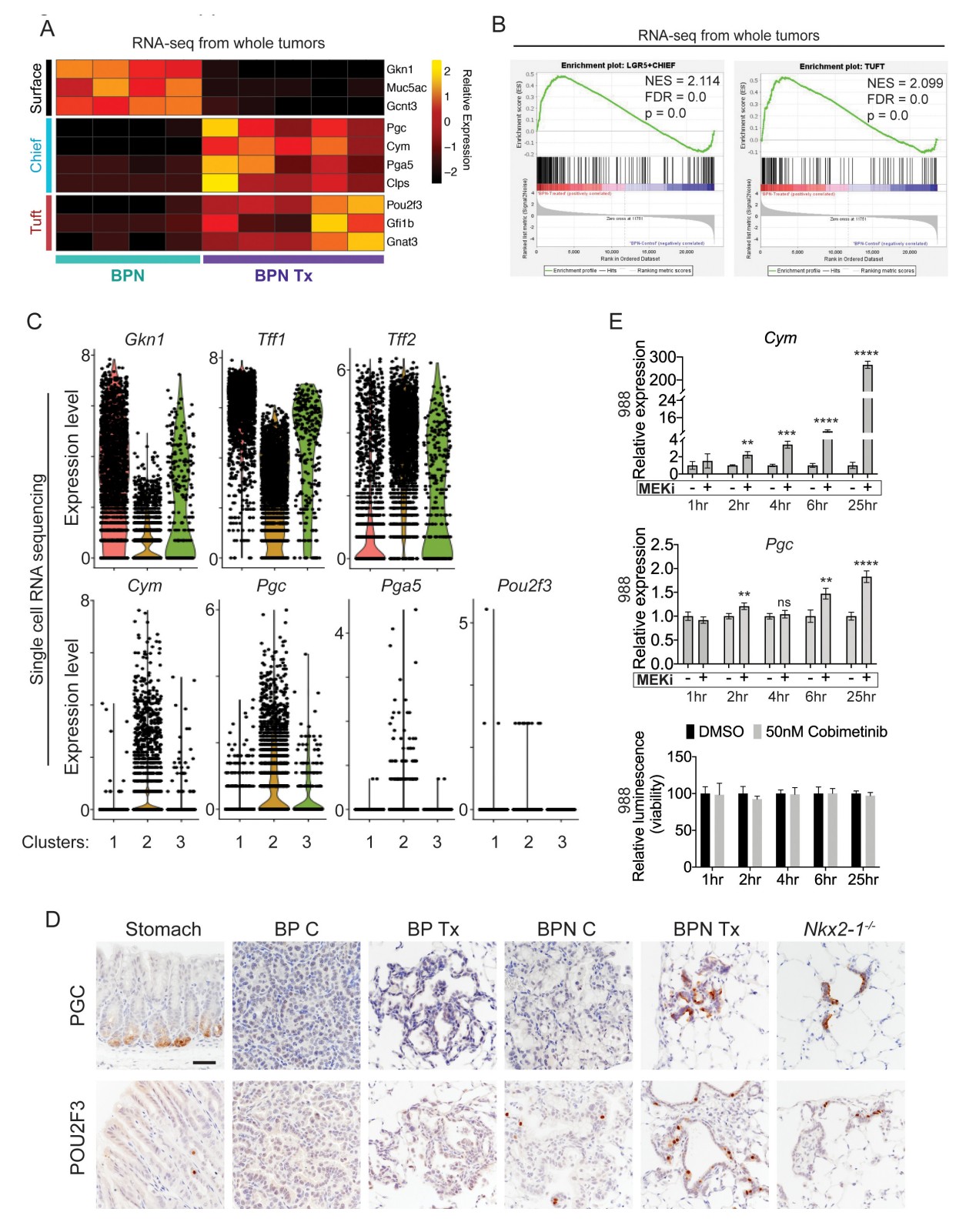

**Figure 6.** MAPK activity regulates cellular identity in NKX2-1-negative tumors. (**A**) Transcriptome analysis comparing control BPN cells with MAPKi-treated BPN tumor cells. Heatmap depicts gastric surface mucous cell markers whose expression diminished versus gastric chief cell or tuft cell markers whose expression increased in MAPK-inhibitor-treated BPN cells relative to untreated. *adjP* <0.05 for each comparison. RNA-seq data was derived from tdTomato-expressing tumor cells FACS-sorted from BPN mice. Six weeks after tumor initiation, mice were treated for 1 week with control chow (n = 4)

*Figure 6 continued on next page*

*Figure 6 continued*

or chow containing BRAF$^{V600E}$ inhibitor plus MEK inhibitor (n = 5). (**B**) GSEA against chief and tuft cell signatures. (**C**) Violin plots of single-cell sequencing data showing expression levels of *Gkn1*, *Tff1*, *Tff2*, *Cym*, *Pgc*, *Pga5*, and *Pou2f3* transcripts in BPN tumor clusters 1–3 and highlighting key drug-mediated cell-identity changes. (**D**) IHC for a gastric chief cell marker (Pepsinogen C) and a tuft cell marker (POU2F3) on LUAD sections derived 7 weeks following tumor initiation with PGK-Cre lentivirus (5 × 10$^3$ pfu/mouse). At 6 weeks, *Braf*$^{LSL-V600E/+}$;*Trp53*$^{f/f}$;*Nkx2-1*$^{f/+}$;*Rosa26*$^{LSL-tdTomato/LSL-tdTomato}$ and *Braf*$^{LSL-V600E/+}$;*Trp53*$^{f/f}$;*Nkx2-1*$^{f/f}$;*Rosa26*$^{LSL-tdTomato/LSL-tdTomato}$ mice were given control or MAPK-inhibitor chow for 1 week. Also shown are sections of normal stomach epithelium as positive controls. Note that in hyperplasia that arise from *Nkx2-1* deletion in distal lung without concomitant oncogenic activation also upregulate these lineage-restricted markers. *Nkx2-1*$^{f/f}$ mice were infected with Ad5-Spc-Cre (10$^9$ pfu/mouse). Lungs were harvested 16 weeks later. Scale bar: 100 µm. (**E**) Chief cell markers are rapidly induced in tumor organoids treated with 50 nM Cobimetinib. qRT-PCR analysis of chief cell markers *Cym* and *Pgc* on RNA isolated from drug/DMSO-treated organoids for the indicated times. Data are mean ± S.D. ****p<0.0001, ***p<0.001, **p<0.01, ns = not significant by Student's *t*-test. A representative experiment of three is shown. Bottom plot: Measurement of viability in organoid cultures treated with vehicle or Cobimetinib using 3D CellTiter-Glo Luminescent cell viability assay at the indicated timepoints. Data are mean ± S.D. A representative experiment of three is shown.

The online version of this article includes the following figure supplement(s) for figure 6:

**Figure supplement 1.** MAPK activity regulates cellular identity in NKX2-1-negative tumors.
**Figure supplement 2.** Durability of BRAF/MEK inhibitor-induced cell identity changes in NKX2-1-negative tumors.

tumors (*Figure 6A*). *Cym* is expressed predominantly during the neonatal phase of mouse stomach development (*Chen et al., 2001*; *Fernandez Vallone et al., 2016*), suggesting that these MAPKi-treated tumor cells may adopt an immature chief cell-like phenotype. MAPK inhibition in NKX2-1-negative tumor cells was also associated with induction of several markers of chemosensory tuft cells, including the lineage specifier *Pou2f3* (*Kaji and Kaunitz, 2017*) as well as *Gfi1b* and *Gnat3* (*Figure 6A*). We next utilized gene set enrichment analysis (GSEA) to compare the transcriptome of drug treated BPN tumors to published/curated gene expression datasets for gastric epithelial cell types and tuft cells (*Haber et al., 2017*; *Leushacke et al., 2017*; *Montoro et al., 2018*; *Ting and von Moltke, 2019*; *Zhang et al., 2019*; *Supplementary file 7, 8*). GSEA showed that signatures of both *Lgr5*-expressing gastric chief cells as well as tuft cells were significantly enriched in MAPK-inhibited BPN cells (*Figure 6B*).

A shift in expression of gastric lineage markers was also evident at the single-cell level. Our single-cell transcriptomics confirmed that gastric surface mucous markers like *Gkn1* and *Tff1* are down-regulated in NKX2-1-negative cells by BRAF/MEK inhibitors (*Figure 6C*), whereas chief markers were induced. Many BRAF/MEK inhibitor-treated BPN cells have higher levels of chief markers (*Cym*, *Pgc*, *Pga5*) than the control BPN cells. This suggests true induction of transcription rather than simple selection for a pre-existing population of chief-high cells (*Figure 6C*). In contrast, MAPK inhibition appears to increase the fraction of cells positive for the tuft cell marker *Pou2f3*, rather than the absolute levels of transcript per cell (though expression of this marker is low and somewhat sporadically detected in our single cell RNA-seq data) (*Figure 6C*). *Tff2* transcripts, which mark progenitors located in the isthmus of the normal stomach (*Quante et al., 2010*), were also increased in BPN cells after MAPKi treatment (*Figure 6C*).

Given this apparent lineage switch, we used IHC to evaluate a subset of lineage markers at the protein level in vivo. GKN1 protein (surface mucous marker) was abundant in control BPN tumors, but was depleted in MAPK inhibitor-treated BPN tumors (*Figure 6—figure supplement 1A*). Alcian Blue staining for mucin production showed the same pattern as GKN1 protein. Control BPN tumors were entirely negative for Pepsinogen C, whereas a subset of drug-treated BPN cells were Pepsinogen C-positive (*Figure 6D*, upper row). In contrast, a small subset of control BPN tumor cells were POU2F3 positive, and the fraction of positive cells was higher upon MAPK inhibition (*Figure 6D*, lower row). These data are concordant with our single-cell analysis of these two markers in terms of changes in proportion of positive cells vs. induction of absolute levels on a per-cell basis. Both Pepsinogen C and POU2F3-positive cells were found in alveolar hyperplasia induced by *Nkx2-1* deletion, consistent with the microscopic similarity between these hyperplasia and residual MAPKi-treated BPN tumor cells. As expected, all three gastric lineage markers evaluated (GKN1, Pepsinogen C and POU2F3) were undetectable in BP tumors. We also analyzed the durability of cell identity changes in BPN induced by BRAF/MEK inhibitors. IHC analysis demonstrated that GKN1 and mucin production (Alcian Blue) return to the same levels as controls within 2 weeks of drug-withdrawal after a month

of treatment with BRAF/MEK inhibitors (*Figure 6—figure supplement 2*). At this timepoint, PGC becomes undetectable and the percentage of POU2F3-positive cells is similar to controls. Thus, based on these markers of major cell state, we conclude that BPN cells readily transition back to their original identity after drug withdrawal.

We also examined POU2F3 by IHC in a panel of primary human LUAD tissues (*Figure 6—figure supplement 1B*). We found that almost all IMA tumors (15/16) harbor a minority population of POU2F3-positive cells (*Figure 6—figure supplement 1B*). Like in BPN tumors, these POU2F3-positive cells were relatively rare (~5% or less of tumor cells overall). In contrast, most NKX2-1-positive human lung adenocarcinomas were entirely POU2F3-negative (44/51) (*Figure 6—figure supplement 1B,C*). Interestingly, we detected a minor population of POU2F3-positive tumor cells in seven NKX2-1-positive cases (*Figure 6—figure supplement 1B*). This expands upon other recent observations that POU2F3 can be upregulated in specific subtypes of lung cancer (*Huang et al., 2018*). Thus, a rare population of POU2F3-positive tumors cells is readily detectable in both human IMA and murine models.

We next asked whether regulation of gastric lineage by the MAPK pathway is a general feature of IMA, or limited to BRAF$^{V600E}$-driven tumors. To address this question, we generated organoids from autochthonous IMA models driven by KRAS$^{G12D}$ (KPN) or BRAF$^{V600E}$ (BPN). IHC for the tdTomato reporter on sections of primary organoid cultures (*Figure 6—figure supplement 1D*) showed that all cultures examined are ~90–100% positive for tdTomato. Rare tdTomato-negative organoids have an exclusively epithelial morphology, and thus are unlikely to represent stromal cell contamination in these cultures. We treated BPN and KPN organoids with a MEK inhibitor (Cobimetinib) for 3 days and found that this was sufficient to stimulate the expression of chief (*Cym*, *Gif*, *Pgc*) and tuft (*Pou2f3* and *Gfi1b*) cell markers in both KPN and BPN organoid cultures (*Figure 6—figure supplement 1E,F*). Furthermore, induction of chief cell markers in BPN organoids occurred rapidly following MEK inhibition (2 hr for *Cym* and 6 hr for *Pgc*), before any change in cell number occurred (*Figure 6E*). Thus, the kinetics of *Cym* induction in vitro are more consistent with a true lineage switch rather than selection for a pre-existing *Cym*-high cells, and are also consistent with our interpretation of single-cell data from tumors in vivo. Finally, we treated BPN organoids with additional drugs that target MAPK signaling including the ERK-inhibitor GDC-0994, MEK inhibitor PD0325901, and the BRAF inhibitor PLX4720, all of which induce *Cym* (*Figure 6—figure supplement 1G*). In aggregate, our data show that drug-induced tumor phenotypic changes are not dependent on the non-specific effects of any one inhibitor. Thus, the MAPK pathway modulates gastric lineage in IMA cells driven by distinct oncogenes (KRAS and BRAF) and in distinct settings (i.e. in vivo and in vitro).

## MAPK inhibition leads to an increase in canonical WNT activity in NKX2-1 negative lung adenocarcinoma

LGR5 marks a subset of normal murine chief cells that can function as facultative stem cells (*Leushacke et al., 2017*). *Lgr5* is also a canonical WNT target gene, and WNT signaling is thought to promote the regenerative capacity of LGR5-positive chief cells. Given that MAPK inhibitors led to an enrichment for the signature of LGR5-positive chief cells in BPN mice, we evaluated our gene expression data to determine whether this was indicative of a general increase in WNT signaling.

We examined a set of WNT pathway genes derived from the 'WNT signaling' ontology category on AmiGO (*Carbon et al., 2009*) in our whole-tumor RNA sequencing of BP and BPN tumors. Several of these genes have previously been identified as direct transcriptional targets of canonical WNT signaling, including *Axin2*, *Lgr5*, *Wnt11*, *Fzd*, *Fgf9*, *Notum*, *Sox9*, *Lrp1*, *Sox4*, *Znrf3*, *Wnt3a* ('positive regulation of canonical WNT signaling' category on AmiGO, *Carbon et al., 2009*) and were significantly activated in MAPK-inhibited BPN tumors (*Figure 7A*). Indeed, applying Ingenuity Pathway Analysis (IPA) Upstream Regulator detection to RNA-seq data from whole tumors identified β-catenin as the second top upstream regulator underlying the gene expression changes between control and drug-treated BPN tumors (*Figure 7—figure supplement 1A*). The activation z-score suggests an activated state of β-catenin due to MAPK-inhibition and a more inhibited state in control BPN tumors. Consistent with this, Regulatory Motif analysis in Illumina Correlation Engine uncovered significant differences in TCF1 binding site genesets between control and MAPKi-treated BPN tumors (*Figure 7—figure supplement 1B*). WNT targets such as *Axin2* (*Jho et al., 2002*) and *Lgr5* (*Barker et al., 2007*) were also induced in our single-cell analysis (*Figure 7—figure supplement*

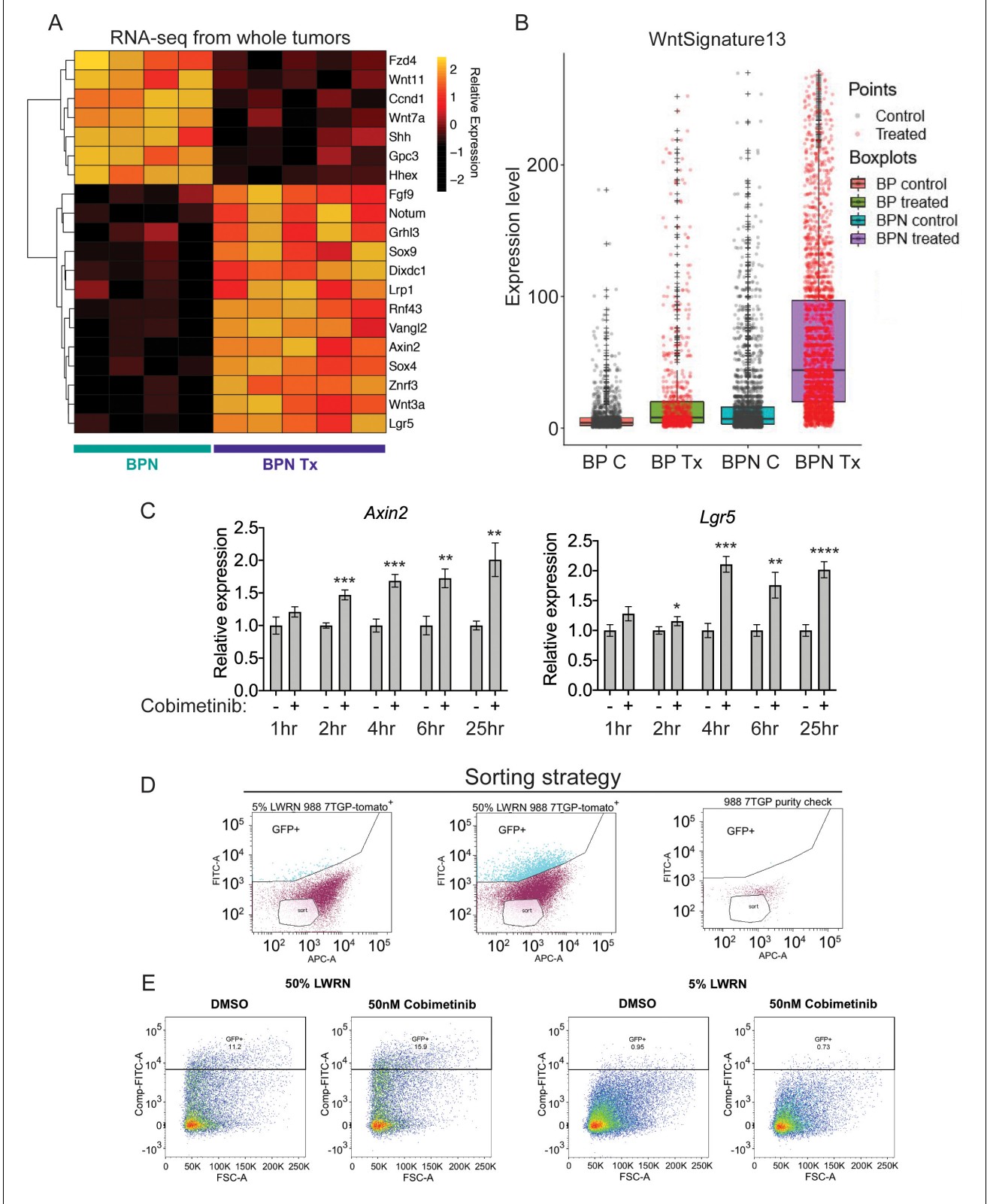

**Figure 7.** MAPK inhibition activates WNT signaling in NKX2-1-negative tumors. (**A**) Transcriptome analysis of genes comprising the canonical WNT pathway gene ontology (AmiGO) in RNA purified from FACS-sorted BPN tumor cells 1 week post- treatment with PLX4720+PD0325901 or control chow. *adjP* <0.05 for each comparison. Color key indicates normalized expression levels (Log10). (**B**) Analysis of WNT pathway activity in scRNA-seq data using a WNT activation signature. BP indicates cells from clusters 4–8; BPN indicates cells from clusters 1–3 (see *Figure 4E, F*). C : control, Tx = drug

*Figure 7 continued on next page*

Figure 7 continued

treatment. (C) Kinetics of WNT signaling induction following MAPK-inhibitor treatment as indicated by qRT-PCR analysis of the canonical WNT signaling markers, *Axin2* and *Lgr5*, on RNA isolated from drug/DMSO treated organoids for the indicated times. Data are mean ± S.D. A representative experiment of three is shown. ****$p<0.0001$, ***$p<0.001$, **$p<0.01$, *$p<0.05$ by Student's *t*-test. (D, E) WNT-low cells can give rise to WNT-high cells. BPN organoid lines, 988 shown here, were stably transduced with a WNT-reporter construct (7TGP, Addgene Plasmid #24305) and FACS-sorted to isolate cells with undetectable WNT activity (pink gate) (D). (E) Dynamics of sorted WNT-low cells when cultured in WNT rich spheroid media (50% LWRN vs. 5% LWRN) or in the presence of MEK-inhibitor. Numbers shown are the percentage of live tdTomato$^+$GFP$^+$ cells in each culture as quantified by flow cytometry (FlowJo). Data shown is a representative of two independent sorting experiments.

The online version of this article includes the following figure supplement(s) for figure 7:

**Figure supplement 1.** MAPK inhibition activates WNT signaling in NKX2-1-negative tumors.

**Figure supplement 2.** Multiple sources for WNT ligands in BRAF$^{V600E}$-driven lung adenocarcinoma.

*1C*). Although their raw values are at the detection limit in our scRNA-seq data, imputation analysis to recover false negatives further supports a robust increase in *Axin2* and *Lgr5* levels after BRAF/MEK inhibition (*Figure 7—figure supplement 1D*).

Intriguingly, the pattern of induction was distinct for these two WNT target genes (*Figure 7—figure supplement 1C,D*). Cells with the highest levels of *Lgr5* are found only in treated BPN tumors, a pattern consistent with our observations of chief cell markers (*Figure 6C*). In contrast, we identified cells with high levels of *Axin2* in both control and treated BPN groups, despite the fact that overall *Axin2* levels are higher in treated BPN. This raises the question of whether WNT-high cells pre-exist in control BPN tumors, or whether MAPK inhibition leads to induction of WNT target genes in WNT-low cells. We therefore developed a WNT13 transcriptional signature (see Materials and methods for derivation) that integrates the WNT target genes induced by MAPK inhibitors in BPN tumors. Scoring of cells in scRNA-seq using this WNT13 signature showed a pattern more similar to *Axin2* than *Lgr5* (*Figure 7B*). Taken together, these data suggest that WNT-high cells can be detected in control BPN tumors, at least as defined by a WNT13 signature or *Axin2* levels. However, *Lgr5*-high cells are not detectable pre-treatment, showing that even WNT-high cells lack high level expression of certain target genes. This further supports our observations from analysis of gastric lineage markers that MAPK inhibition modulates the transcriptional cell state of BPN tumor cells (including WNT target genes), and does not purely select for pre-existing populations.

To further investigate this question, we performed in vitro time course experiments, which showed that MEK inhibition induces *Lgr5* and *Axin2* within 4 hr of treatment before changes in cell number or viability take place (*Figures 7C* and *6E*). In a different approach to study the emergence of WNT-high cells, we stably transduced a BPN organoid line with a WNT-reporter construct, 7TGP, that contains seven TCF binding sites driving eGFP expression (*Figure 7D,E*). When cultured in standard WNT-pathway activating 50% L-WRN media, we found that these cells consisted of a mixture of high and low reporting cells (WNT$_{rep}$-high and WNT$_{rep}$-low cells respectively). In contrast, most cells were WNT$_{rep}$-low when cultured in 5% L-WRN media. We then sorted the WNT$_{rep}$-low subpopulation from standard culture conditions (50% L-WRN media) and found that these cells re-equilibrated to a mix of WNT$_{rep}$-high and WNT$_{rep}$-low cells when cultured in 50% L-WRN media (*Figure 7E*). Furthermore, culture of the sorted cells in 5% L-WRN media reduced the WNT$_{rep}$-high percentage, whereas MEK inhibitor increased the WNT$_{rep}$-high percentage. Taken together, these experiments indicate that WNT signaling can be induced in WNT$_{rep}$-low cells with rapid kinetics (i.e. prior to a change in cell viability or selection). Although we cannot exclude the possibility that WNT$_{rep}$-high cells tolerate MEK inhibition to a greater extent than WNT$_{rep}$-low cells, these data argue that pure selection for a WNT$_{rep}$-high subpopulation (in the absence of changes in gene expression on a per cell basis) is unlikely to account for our observations on the overall state change of tumors.

Previous studies in other cancer types have documented negative feedback loops between the MAPK and WNT signaling pathways. Mechanistically, these have been reported to be mediated by changes in AXIN1 (*Zhan et al., 2019*), TCF4 isoform levels (*Heuberger et al., 2014*) or phosphorylation of FAK (*Chen et al., 2018*) after MAPK inhibition. However, we have not been able to detect

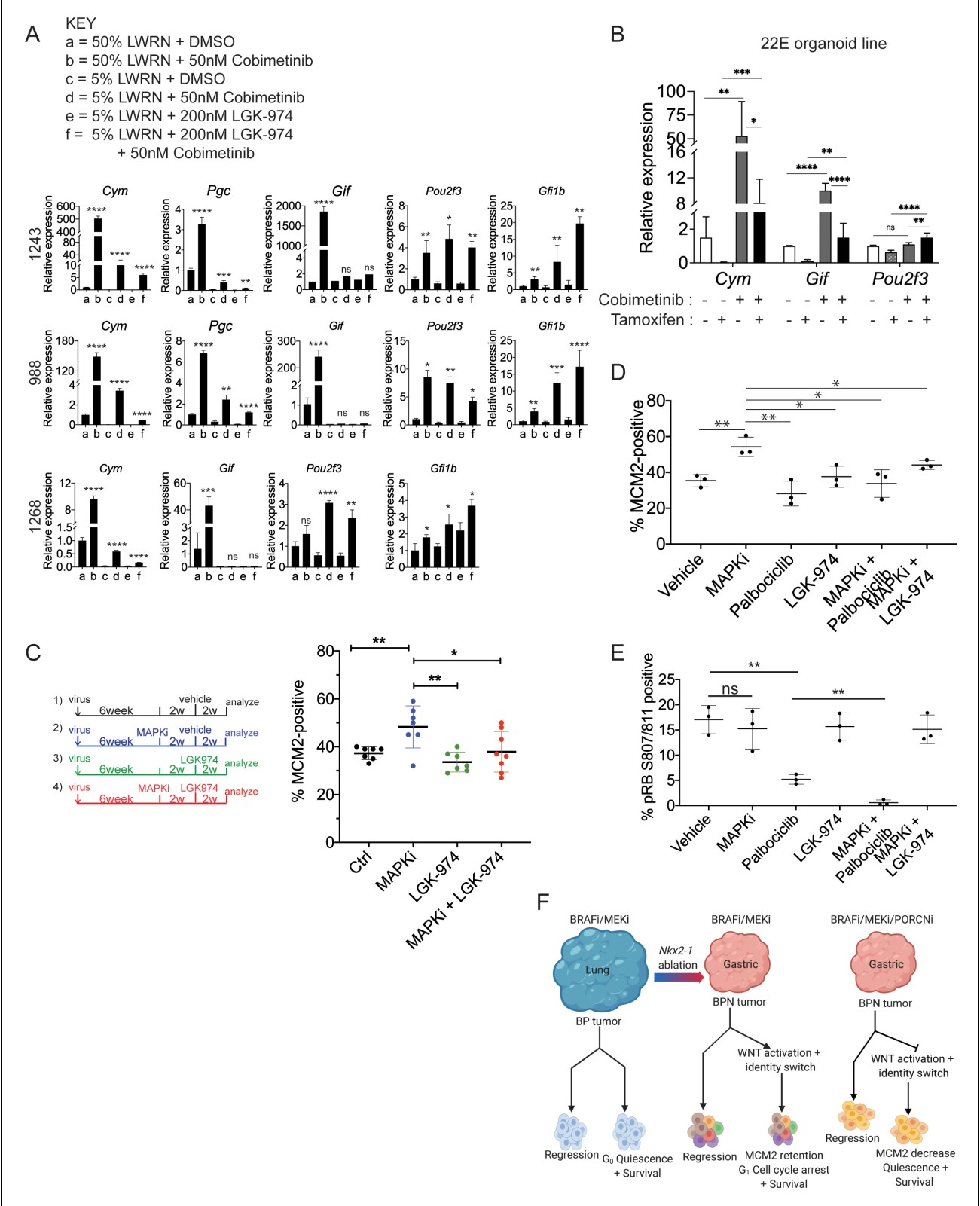

**Figure 8.** WNT signaling and the transcription factors FoxA1/FoxA2 are partially required for lineage switching induced by MAPK inhibition. (**A**) Analyses of indicated gene expression levels in BPN (1243 and 988) and KPN (1268) tumor organoid lines by qRT-PCR at 24 hr and under different treatment conditions. Organoids were cultured in 50% L-WRN (**a, b**) or reduced 5% L-WRN media (**c, d, e, f**) and treated with DMSO (**a, c**), single agent Cobimetinib (**b, d**) or the Porcupine inhibitor, LGK-974 (**e**), and both inhibitors (**f**). Graphs indicate mean ± S.D. [p values are for *a-b; c-d;* or *e-f*

*Figure 8 continued on next page*

Figure 8 continued

comparisons]. Experiment was reproducibly performed three times. (**B**) qRT-PCR analysis of the expression levels of indicated cell identity markers in the 22E organoid line under four different conditions. 22E (derived from a lung tumor in a $Kras^{FSF\text{-}G12D/+}$;$Trp53^{frt/frt+}$;$Foxa1^{f/f}$; $Foxa2^{f/f}$;$Rosa26^{FSF\text{-}CreERT2/}$ $^{FSF\text{-}CreERT2}$ mouse) lacks NKX2-1 expression and harbors conditional alleles of $Foxa1$ and $Foxa2$. Tamoxifen treatment induces Cre$^{ERT2}$-mediated $Foxa1/Foxa2$ deletion. Data are mean ± S.D and a pool of two independent experiments. (**C**) Quantitation of the proliferation marker MCM2 in 10 week autochthonous BPN lung tumors under four different treatment conditions: (1) control (n = 7); (2) MAPK-inhibitor chow (n = 7); (3) LGK-974 (n = 7); and (4) MAPK-inhibitor chow and LGK-974 (n = 8). (**D, E**) Abundance of MCM2 or phospho-RB-positive cells in 10-week autochthonous lung tumors from BPN mice under the indicated conditions. Vehicle or drug treatments were administered at 6 weeks post tumor initiation for 4 weeks. Multiple tumors from n = 3 mice/group were quantitated. (**A – E**) ****p<0.0001, ***p<0.001, **p<0.01, *p<0.05, ns = not significant by Student's $t$-test. (**F**) Graphical depiction of the molecular and pharmacological regulation of differentiation programs in LUAD (BioRender). Here, we investigated genotype-specific drug response. Cancer therapy triggers two alternative fates in drug-treated tumors of both genotypes: either regression or survival in a drug tolerant state. Molecularly, the drug tolerant state appears distinct in BP versus BPN tumors and can be exploited pharmacologically by targeting the cell cycle and/or drug-induced signaling pathways.

The online version of this article includes the following figure supplement(s) for figure 8:

**Figure supplement 1.** WNT signaling and the transcription factors FoxA1/FoxA2 are partially required for lineage switching induced by MAPK inhibition.

**Figure supplement 2.** Characterization of MAPK inhibitor-driven signaling changes.

consistent changes in any of these parameters after treatment of our organoids with MEK inhibitors (*Figure 7—figure supplement 1E*).

To identify potential sources of WNT production, we analyzed the levels of *Wnt* transcripts in all cell clusters identified in scRNA-seq, including stromal cells (see *Figure 4—figure supplement 1E*). As presented in *Figure 7—figure supplement 2*, several *Wnt* genes were induced by MAPKi treatment in BPN tumor cells, including *Wnt2b*, *Wnt4*, *Wnt8b*, and *Wnt10b*. Additional *Wnt* ligands such as *Wnt2*, *Wnt5b* and *Wnt6* were more highly expressed in stromal cells that were co-isolated during lung tumor preparation. Thus, there are multiple potential in vivo sources of WNT ligand, including both treated BPN tumor cells and the tumor stroma.

## WNT signaling and FoxA1/2 contribute to lineage switching and cell cycle response in NKX2-1-negative lung adenocarcinoma treated with RAF/MEK inhibitor

WNT/β-catenin signaling plays an essential role in gastric epithelial patterning during embryogenesis and promotes chief cell differentiation (*McCracken et al., 2017*). We therefore asked whether the WNT pathway played a role in the lineage switch induced by MAPK inhibition in IMA. We used two different approaches to modulate WNT signaling in organoid cultures. First, we cultured organoids in 5% conditioned media (L-WRN), corresponding to a 10 fold reduction in the amount of exogenous WNT3A and R-spondin three in the culture media relative to standard, 50% L-WRN, conditions. Second, we used the small molecule LGK-974, which blocks Porcupine-mediated posttranslational modification of WNT ligands that is required for their secretion and paracrine signaling. Levels of *Axin2* and *Lgr5* were lower in organoids cultured in 5% L-WRN than 50% L-WRN, and treatment with LGK-974 led to a further reduction in these transcripts suggesting endogenous WNT ligand production and signaling complements L-WRN media in these cultures (*Figure 8—figure supplement 1A*). Consistent with prior analyses (*Figure 7C*), MEK inhibitor stimulated canonical WNT signaling in BPN and KPN organoids as indicated by increased expression of the *Axin2* and *Lgr5* transcripts within the first 24 hr (*Figure 8—figure supplement 1A*). The relative effect of this stimulation (i.e. fold induction) was similar whether the organoid media composition contained high or low levels of exogenous WNT3A. Induction of chief markers (*Cym*, *Pgc*, *Gif*) by Cobimetinib was partially impaired when WNT signaling was reduced by using 5% L-WRN media alone or in combination with LGK-974 (*Figure 8A*). Levels of some chief markers (e.g. *Cym*) were responsive to WNT signaling levels even in the absence of MEK inhibitor. In contrast, tuft cell markers (*Pou2f3* and *Gfi1b*) were largely unaffected by modulation of WNT signaling. These data show that one component of lineage switching (induction of chief cell markers) is partially WNT-dependent, whereas another component (tuft cell marker induction) is largely WNT-independent.

Previously, our lab reported that FoxA1 and FoxA2 are required for gastric differentiation in mouse models of IMA, including surface mucous cell marker expression (*Camolotto et al., 2018*). Here, using a novel NKX2-1-negative organoid line that harbors conditional alleles of *Foxa1* and *Foxa2*, we show that *Foxa1/2* deletion abrogates the induction of chief cell markers (but not tuft cell markers) upon MEK inhibition (*Figure 8B*, *Figure 8—figure supplement 1B*). Thus, the surface to chief gastric lineage switch is driven by both WNT signaling and FoxA1/2 activity.

Finally, we asked whether WNT signaling might play a role in failure of BPN tumors to exit the cell cycle after RAF/MEK inhibition. WNT signaling contributes to oncogenesis in a variety of settings (*Zhan et al., 2017*), including in NKX2-1-positive LUAD GEMMs driven by BRAF^V600E (*Juan et al., 2014*) or KRAS^G12D (*Tammela et al., 2017*). Further, β-catenin activation upon *Apc* deletion in the murine intestinal epithelium specifically upregulates cyclin D2 to augment proliferation and tumorigenesis (*Cole et al., 2010*). This raises the possibility that activation of the WNT pathway might be sufficient to induce *Ccnd2* in MAPKi-treated BPN tumors and thereby prevent cell cycle exit, despite the lack of MAPK activity.

To investigate this possibility, we first correlated known signatures of MEK (*Dry et al., 2010*) and WNT activity ('positive regulation of canonical WNT signaling' category on AmiGO, *Carbon et al., 2009*) with levels of *Ccnd1* and *Ccnd2* in our single-cell data. MEK activity was positively correlated to *Ccnd1* levels across the single-cell data set. Conversely, *Ccnd2* levels were inversely correlated to the MEK signature score and positively correlated to the signature for WNT activity (*Figure 8—figure supplement 1C*). This suggests that *Ccnd1* levels may be primarily dependent on MAPK activity, whereas *Ccnd2* levels may be more directly dependent on WNT activity in this specific context.

Given the relationship between WNT activity and *Ccnd2* levels, we asked whether pharmacological WNT inhibition would be sufficient to drive RAF/MEK-inhibited BPN cells into quiescence. To address this question, we initiated lung tumors in a cohort of BPN mice and assigned mice to four groups at 6 weeks after initiation (vehicle; MAPKi; LGK-974; or MAPKi + LGK-974). MAPK inhibitor was administered for 4 weeks (i.e. 6–10 weeks after tumor initiation) and LGK-974 was administered for the last 2 weeks of MAPKi treatment (i.e. 8–10 weeks after tumor initiation). Four weeks of treatment with RAF/MEK inhibitor led to a significant increase in the percentage of MCM2-positive cells (*Figure 8C*), consistent with the effects we observed after 1–2 weeks of treatment. In contrast, treatment with LGK-974 alone had no effect on MCM2 in BPN tumors. Strikingly, additional treatment with LGK-974 was sufficient to block the induction of MCM2 by RAF/MEK inhibition in BPN tumors, resulting in an MCM2-positive rate similar to controls (*Figure 8C*). This demonstrates that induction of WNT signaling after RAF/MEK inhibition acts to prevent residual BPN tumor cells from entering quiescence. Although the addition of short-term LGK-974 to RAF/MEK inhibition did not reduce tumor burden beyond RAF/MEK inhibition alone (*Figure 8—figure supplement 1D*), our data show that the biology of residual tumor cells can be manipulated by modulating WNT signaling.

We also assessed the status of β-catenin in drug-treated BPN tumors in vivo. Using an antibody that recognizes non-phosphorylated (active) β-catenin or a second antibody for total β-catenin (*Figure 8—figure supplement 2A,B*), we found that vehicle and LGK-974-treated tumors exhibit predominantly membranous staining by IHC, with no evidence of nuclear β-catenin. MAPK inhibition alone elicited accumulation of β-catenin throughout the cell, including the nucleus. Levels of nuclear/cytoplasmic β-catenin in tumors diminished in the presence of combined MAPK and LGK-974. These findings are consistent with the possibility that β-catenin mediates MAPK-inhibition-induced WNT pathway activation and lineage switching in BPN tumors. In contrast, control BP tumors had lower levels of active β-catenin than BPN tumors, and there was no appreciable increase in staining with MAPK inhibition.

Finally, we performed a larger experiment in which we combined RAF/MEK inhibition with either WNT inhibition (LGK-974) or CDK4/6 inhibition (Palbociclib) for 4 weeks. We analyzed a subset of mice at the end of 4 weeks of drug treatment and performed a survival study on the remaining mice. In mice analyzed immediately after the final dose of drug, we found that adding either LGK-974 or Palbociclib to RAF/MEK inhibitors increased the number of cells in quiescence compared to RAF/MEK inhibitor alone (*Figure 8D*). Addition of Palbociclib to the RAF/MEK inhibitor also reduced the amount of phospho-RB (S807/811) in the tumors (*Figure 8E*). These data suggest a model in which elevated WNT signaling in MAPKi-treated BPN tumors prevents cell cycle exit by maintaining higher levels of CDK4/6 activity than in BP tumors treated with the same drug. Our gene expression data

suggest that Cyclin D2 is the CDK4/6 partner that is most likely to maintain its activity in MAPKi-treated BPN tumors.

Despite the short term effects of adding a third drug, we did not see a significant increase in survival when either LGK-974 or Palbociclib was added to RAF/MEK inhibitor (*Figure 8—figure supplement 1E*). This indicates that driving residual BPN cells out of the cell cycle is not sufficient to prevent tumor rebound after drug cessation.

## Discussion

LUAD progression is driven not only by activation of MAPK signaling but also by changes in cellular identity. However, the direct impact of dysregulated MAPK activity on LUAD identity itself remains an unexplored problem in lung cancer. This problem is important not only because of the correlation between cellular identity and intrinsic malignant potential, but also because lineage switching has become an increasingly recognized, common mechanism of resistance to therapies targeting the MAPK pathway. A better understanding of this phenomenon would provide new insights into the natural history of LUAD progression as well as the changes in cellular identity that occur as a result of targeted therapy.

Analyses of the histologic spectrum of LUAD in patients and mouse models have identified two cellular differentiation programs that impose barriers to lung tumor progression (*Camolotto et al., 2018*; *Snyder et al., 2013*; *Winslow et al., 2011*). Initially, NKX2-1/FOXA1/2 transcriptional networks maintain a well-differentiated pulmonary identity. We have previously shown that downregulation of NKX2-1 and subsequent relocalization of FOXA1 and FOXA2 to the regulatory elements of gastrointestinal lineage genes allows for loss of surfactant proteins, aberrant upregulation of HNF4A and conversion to gastric cell identity. Loss of both differentiation programs can lead to poorly differentiated tumors containing high levels of HMGA2.

Human IMAs are associated with adverse clinical outcomes and harbor a distinct spectrum of driver mutations. Compared to LUAD overall, *KRAS* mutations are much more common in IMA, whereas *BRAF* mutations are somewhat less frequent (*Cha and Shim, 2017*). There is emerging evidence that tumor suppressor gene mutations may also be distinct in IMA. For example, *CDKN2A* mutations appear to be more common than *TP53* mutations in human IMA (*Shim et al., 2015*; *Skoulidis et al., 2015*). Intriguingly, a case report demonstrated that *TP53* mutation correlated with progression of a human IMA to a higher grade adenocarcinoma with decreased mucin production (*Kawai et al., 2019*). This suggests that *TP53* mutations can occur in human IMA but can facilitate progression to a non-mucinous differentiation state. This mirrors what we have observed in BPN and KPN models, and further supports their clinical relevance.

GEMMs have been valuable in explaining the distinct role of NKX2-1 in LUAD driven by *KRAS* and *EGFR* mutations (*Maeda et al., 2012*; *Snyder et al., 2013*). In the present study, we show that NKX2-1 is required for optimal LUAD initiation by BRAF$^{V600E}$, but it is dispensable for growth and progression to high grade states in established tumors. We speculate that concomitant BRAF$^{V600E}$ activation and *Nkx2-1* deletion leads to a high level of ERK activity that is not well tolerated by normal cells. BRAF$^{V600E}$-driven neoplastic cells, which have already equilibrated to moderately increased ERK activity, might more readily tolerate a further increase in ERK activity caused by *Nkx2-1* deletion. In contrast to our results in the BRAF$^{V600E}$ models, loss of NKX2-1 consistently promotes tumorigenesis in KRAS$^{G12D}$-driven GEMMs. Taken together, these data explain why *KRAS* mutations are enriched in IMA, whereas *BRAF* mutations are not. It remains to be determined whether co-mutations can render NKX2-1 loss advantageous (rather than neutral) in *BRAF*-mutant LUADs. For example, activating codon 201 mutations in *GNAS*, which are predicted to drive cAMP-mediated Protein Kinase A (PKA) activity, occur more frequently in IMA than other LUAD subtypes (*Ritterhouse et al., 2017*). Although functional studies of GNAS mutations in IMA have not been reported, codon 201 GNAS mutants function as oncogenes in KRAS-driven GEMMs of pancreatic cancer, which expresses some of the same foregut markers as IMA (*Patra et al., 2018*; *Taki et al., 2016*). In our RNA sequencing, several mRNAs encoding negative regulators of cAMP-mediated signaling to PKA, including *Prkar2b*, *Rgs7*, and *Adra2a* (*Garg et al., 2016*; *Ghavami et al., 2004*; *Taylor et al., 2008*), were significantly upregulated in BPN cells compared to BP. Interestingly, these transcripts were not elevated in *Nkx2-1* deleted

KRAS-driven tumor cells (*Camolotto et al., 2018*) and decreased cAMP-dependent PKA activity may be one reason why tumor initiation is impaired in BPN tumors.

We have previously shown that the gastric differentiation state of IMA is driven by transcription factors such as FoxA1/2 and HNF4α. Here, we show that MAPK and WNT signaling provide an additional layer of regulation, modulating the specific cell type that IMA cells most closely resemble within the gastric lineage. Our data reveal that high MAPK activity drives a gene expression program characteristic of the surface mucous cells of the stomach. In contrast, low ERK levels, in concert with high WNT activity, activate gene expression signatures associated with other gastric cell types, including LGR5-positive chief cells and tuft cells. POU2F3-positive tuft-like cells have also been identified in pancreatic neoplasia, where they restrain tumorigenesis (*DelGiorno et al., 2020*). Interestingly, hyperactivation of MAPK signaling (via TGF-α mediated activation of EGFR) is observed in Menetrier's disease, a hypertrophic gastropathy characterized by hyperproliferation of isthmus progenitors and the preferential differentiation of these progenitors into surface mucous cells at the expense of parietal and chief cells (*Fiske et al., 2009*). Blockade of EGFR results in regression of overgrown surface epithelium and restoration of lineage fidelity in progenitors.

Importantly, our results identify crosstalk between the MAPK and WNT pathways in IMA tumors, as suppression of MAPK pathway rapidly activates WNT target gene expression. In the mammalian intestine, reciprocal gradients of WNT and phosphorylated ERK1/2 control the balance between proliferation and differentiation of intestinal stem cells (*Kabiri et al., 2018*). Some molecular mechanisms driving the crosstalk between WNT and ERK signaling have been reported. In one study, reduction of MEK activity via gut-specific ablation of *Ptpn11* decreased abundance of TCF4M/S isoforms while favoring the binding of β-catenin to TCF4E variants, which are more efficient drivers of WNT target gene activation (*Heuberger et al., 2014*). Alternatively, treatment of BRAF mutant colorectal cancer models with MAPK inhibitors stimulated WNT pathway through the activation of cytoplasmic focal adhesion kinase (FAK) (*Chen et al., 2018*) or depletion of AXIN1 protein (*Zhan et al., 2019*), resulting in the stabilization of β-catenin. Although we find no evidence for the above three mechanisms as mediators of ERK/WNT cross-regulation in our lung tumor model, we did identify three candidate genes induced by MAPK inhibition (whole-tumor and single cell RNA-seq) that might account for the increased WNT signaling: *Dixdc1*, *Sox4* and *Sox9* (*Figure 7A*). DIXDC1 (aka CCD1) is a scaffolding protein that promotes assembly of Dishevelled into its biologically active, trimeric, form capable of interacting with AXIN and recruiting the destruction complex to the plasma membrane (*Liu et al., 2011*). SOX4 is a transcription factor that can interact with AXIN-APC destruction complex and interfere with the phosphorylation of β-catenin, thereby increasing its stability (*Bhattaram et al., 2014*). On the other hand, SOX9 can directly transactivate WNT target genes in specific contexts (*Ma et al., 2016*). Our preliminary in vitro data indicates that these transcripts are induced by at least 1.5 fold within one hour of Cobimetinib treatment, preceding the upregulation of *Axin2*/*Lgr5*, and continue to accumulate over the course of 24 hr. However, establishing whether any of these factors are required for the interplay between ERK/WNT signaling axes in drug-treated IMAs requires further experimental investigation.

Our results also have translational implications for LUAD treatment. *BRAF*-mutant LUADs exhibit a heterogenous response to BRAF/MEK inhibition in clinical trials (*Planchard et al., 2016*; *Planchard et al., 2017*). Here, we identify NKX2-1 as a determinant of treatment response in a GEMM of this disease. In both NKX2-1-positive and NKX2-1-negative tumors, combined BRAF/MEK inhibition leads to substantial tumor regression, but drug-tolerant cells persist with the potential to relapse. In the absence of NKX2-1, residual tumor cells fail to exit the cell cycle. The different cell cycle positions of residual BP and BPN tumor cells suggest that different therapeutic approaches will be needed to eradicate these cells. Indeed, we were able to identify factors mediating these differences, including elevated Cyclin D2-CDK4/6 activity and sustained phosphorylation of RB. We verified that the cell cycle position of MAPK-inhibited tumor cells can be manipulated by downregulating Cyclin D2 or blocking CDK4/6 activity via Palbociclib. However, these tripartite drug combination strategies were not efficacious in achieving durable tumor regression or survival benefit. It is interesting to consider these results in the context of two recent studies where the combinatorial effect of Palbociclib and MEK inhibitor (Trametinib) was tested in models of KRAS-driven NKX2-1-positive LUAD (*Ruscetti et al., 2018*) and in KRAS-driven pancreatic ductal adenocarcinoma (PDAC) (*Ruscetti et al., 2020*). Consistent with our findings, Trametinib/Palbociclib co-administration in both types of mice elicited a potent inhibition of RB phosphorylation. In addition, combination

therapy in KRAS-driven lung tumors resulted in prominent induction of cellular senescence and infiltration of tumors with activated NK cells, which drove tumor cell clearance, and markedly increased survival in dual-treatment mice compared to single agent treated mice (*Ruscetti et al., 2018*). In PDAC models, however, dual treatment with Trametinib/Palbociclib elicited tumor senescence but not NK cell recruitment and failed to elicit robust tumor regression (*Ruscetti et al., 2020*). Importantly, BPN tumors have a gastric/foregut-like differentiation state that is much closer to PDAC than to KRAS-driven NKX2-1-positive LUAD. Thus, our results expand upon these published observations that lineage state can influence the response to combined inhibition of MAPK and CDK4/6 activity. It appears that immune cell composition of NKX2-1-positive and NKX2-1-negative LUAD GEMMs can be distinct (*Mollaoglu et al., 2018*). Thus, investigations into the tumor microenvironment of IMAs and how it is impacted by combination therapies may also provide important clues as to the mechanisms that maintain drug tolerant states.

Additional work is clearly needed to identify specific vulnerabilities of residual IMA cells after MAPK inhibition. The lineage switch we observe in both KRAS and BRAF-driven IMA models is likely to be relevant to the identification of such vulnerabilities. MAPK inhibition causes IMA cells to undergo an identity shift from a surface mucous cell-like phenotype toward a more gastric stem cell-like state or tuft cell identity. These observations support the broader notion that tumor cell plasticity includes the capacity to explore myriad cell states from the developmental repertoire. Lineage switching may be particularly beneficial for tumor cells under the selective pressure of targeted therapy, as diverse lineage programs have distinct dependencies on specific growth and survival mechanisms. We speculate that cell identity changes within the gastric lineage may render residual IMA cells susceptible to specific therapeutic interventions. This would be reminiscent of prior studies of basal cell carcinoma of the skin showing that an identity switch confers resistance to inhibition of the Hedgehog pathway while rendering the cells susceptible to blockade of WNT signaling (*Biehs et al., 2018*; *Sánchez-Danés et al., 2018*).

In summary, this study characterizes the context-dependent role of NKX2-1 in LUAD and identifies novel mechanisms of cell identity regulation and therapeutic response in this disease (*Figure 8F*). The data provide new insights into the complex interplay between lineage specifiers, oncogenic signaling pathways and the susceptibility of lung cancer cells to targeted therapy. Thus, in the interest of developing fully effective therapies, these studies call for deep cataloguing of cell states sampled by tumor cells as well as their state-specific vulnerabilities.

# Materials and methods

**Key resources table**

| Reagent type (species) or resource | Designation | Source or reference | Identifiers | Additional information |
|---|---|---|---|---|
| Genetic reagent (*Mus musculus*) | Braf$^{LSL-V600E/+}$ | *Dankort et al., 2007* PMID:17299132 | MGI:3711771 | Dr. Martin McMahon (HCI, Salt Lake City, Utah); mixed C57BL/6J × 129SvJ background |
| Genetic reagent (*Mus musculus*) | Braf$^{FSF-V600E/+}$ | *Shai et al., 2015* PMID:26001956 | | Dr. Martin McMahon (HCI, Salt Lake City, Utah); mixed C57BL/6J × 129SvJ background |
| Genetic reagent (*Mus musculus*) | Trp53$^{f/f}$ | *Jonkers et al., 2001* PMID:11694875 | MGI:1931011 | Dr. Anton Berns, University of Amsterdam; Jackson Laboratories (Bar Harbor, Maine); mixed C57BL/6J × 129SvJ background |
| Genetic reagent (*Mus musculus*) | Trp53$^{frt/frt}$ | *Lee et al., 2012* PMID:22228755 | MGI:5306612 | Dr. David G Kirsch (Duke University Medical Center, Durham, North Carolina); mixed C57BL/6J × 129SvJ background |

*Continued on next page*

*Continued*

| Reagent type (species) or resource | Designation | Source or reference | Identifiers | Additional information |
|---|---|---|---|---|
| Genetic reagent (*Mus musculus*) | Kras$^{LSL-G12D/+}$ | *Jackson et al., 2001* PMID:11751630 | MGI:2429948 | Dr. Tyler Jacks (MIT, Cambridge, Massachusetts); mixed C57BL/6J × 129SvJ background |
| Genetic reagent (*Mus musculus*) | Kras$^{FSF-G12D/+}$ | *Young et al., 2011* PMID:21512139 | MGI:5007794 | Dr. Tyler Jacks (MIT, Cambridge, Massachusetts); mixed C57BL/6J × 129SvJ background |
| Genetic reagent (*Mus musculus*) | Nkx2-1$^{f/f}$ | *Kusakabe et al., 2006* PMID:16601074 | MGI: 3653706 | Dr. Shioko Kimura (NCI, NIH, Bethseda, Maryland); mixed C57BL/6J × 129SvJ background |
| Genetic reagent (*Mus musculus*) | Rosa26$^{LSL-tdTomato}$ Ai14 | *Madisen et al., 2010* PMID:20023653 | MGI: 4436847 | Jackson Laboratories (Bar Harbor, Maine); mixed C57BL/6J × 129SvJ background |
| Genetic reagent (*Mus musculus*) | Rosa26$^{FSF-CreERT2}$ | *Schönhuber et al., 2014* PMID:25326799 | MGI: 5616874 | Dr. Dieter Saur (Technische Universitat Munchen, Munchen, Germany); mixed C57BL/6J × 129SvJ background |
| Genetic reagent (*Mus musculus*) | Foxa1$^{f/f}$ | *Gao et al., 2008* PMID:19141476 | MGI: 3831163 | Dr. Klaus H. Kaestner (Univ. of Pennsylvania School of Medicine, Philadelphia, PA); mixed C57BL/6J × 129SvJ background |
| Genetic reagent (*Mus musculus*) | Foxa2$^{f/f}$ | *Sund et al., 2000* PMID:10866673 | MGI: 2177357 | Dr. Klaus H. Kaestner (Univ. of Pennsylvania School of Medicine, Philadelphia, PA); mixed C57BL/6J × 129SvJ background |
| Genetic reagent (*Mus musculus*) | NOD/SCID-gamma chain deficient (NSG) | The Jackson Laboratory | 005557 | |
| Cell line (*Homo sapiens*) | 293T | *DuPage et al., 2009* PMID:19561589 | | |
| Cell line (*Mus musculus*) | L-WRN | ATCC | CRL-3276 | |
| Recombinant DNA reagent | d8.9 (plasmid) | *DuPage et al., 2009* PMID:19561589 | | |
| Recombinant DNA reagent | VSV-G (plasmid) | *DuPage et al., 2009* PMID:19561589 | | |
| Recombinant DNA reagent | 7TGP (plasmid) | Addgene | 24305 | |
| Chemical compound, drug | Tamoxifen | Sigma Aldrich | T5648 | |
| Chemical compound, drug | Tamoxifen supplemented chow | Envigo | TD.130858 | |
| Chemical compound, drug | PLX4720 supplemented chow | Plexxikon/ Research Diets | | *Tsai et al., 2008* |
| Chemical compound, drug | PD0325901 supplemented chow | Plexxikon/ Research Diets | | *Trejo et al., 2012* |
| Chemical compound, drug | PLX4720 | Selleckchem | S1152 | |
| Chemical compound, drug | PD0325901 | Selleckchem | S1036 | |
| Chemical compound, drug | GDC-0994 | Genentech | | *Blake et al., 2016* |

*Continued on next page*

*Continued*

| Reagent type (species) or resource | Designation | Source or reference | Identifiers | Additional information |
|---|---|---|---|---|
| Chemical compound, drug | Cobimetinib GDC-0973 | Genentech | | |
| Chemical compound, drug | Palbociclib | LC Laboratories | P-7744 | |
| Chemical compound, drug | LGK-974 | Selleckchem | S7143 | |
| Antibody | Anti-NKX2-1 (Rabbit monoclonal) | Abcam | Cat# ab76013 | (1:2000) |
| Antibody | Anti-DUSP6 (Rabbit monoclonal) | Abcam | Cat# ab76310 | (1:500) |
| Antibody | Anti-SPRY2 (Rabbit monoclonal) | Abcam | Cat# ab180527 | (1:1000) |
| Antibody | Anti-pERK (Rabbit monoclonal) | Cell Signaling Technology | Cat# 4370 | (1:500) |
| Antibody | Anti-ERK (Mouse monoclonal) | Cell Signaling Technology | Cat# 4696 | (1:2000) |
| Antibody | Anti-pRSK S380 (Rabbit monoclonal) | Cell Signaling Technology | Cat# 11989 | (1:300) |
| Antibody | Anti-p4EBP1 (Rabbit monoclonal) | Cell Signaling Technology | Cat# 2855 | (1:400) |
| Antibody | Anti-pS6 S235/236 (Rabbit monoclonal) | Cell Signaling Technology | Cat# 4858 | (1:400) |
| Antibody | Anti-pS6 S240/244 (Rabbit monoclonal) | Cell Signaling Technology | Cat# 5364 | (1:1000) |
| Antibody | Anti-pMEK S221 (Rabbit monoclonal) | Cell Signaling Technology | Cat# 2338 | (1:100) |
| Antibody | Anti-PGC (Rabbit polyclonal) | Cell Signaling Technology | Cat# HPA031718 | (1:100) |
| Antibody | Anti-POU2F3 (Rabbit polyclonal) | Cell Signaling Technology | Cat# HPA019652 | (1:200) |
| Antibody | Anti-pRB S807/S811 (Rabbit polyclonal) | Cell Signaling Technology | Cat# 8516 | (1:800) |
| Antibody | Anti-CD36 (Rat monoclonal) | R and D Systems | Cat# MAB25191 | (1:300) |
| Antibody | Anti-proSPB (Rabbit polyclonal) | Millipore | Cat# AB3430 | (1:3000) |
| Antibody | Anti-proSPC (Rabbit polyclonal) | Millipore | Cat# AB3786 | (1:4000) |
| Antibody | Anti-Gastrokine 1 (Mouse monoclonal) | Abnova | Cat# H00056287-M01 | (1:50) |
| Antibody | Anti-Muc5AC (Mouse monoclonal) | Abnova | Cat# MAB13117 | (1:100) |
| Antibody | Anti-Histone H3 phospho-Ser10 (Rabbit polyclonal) | Abcam | Cat# ab5176 | (1:200) |
| Antibody | Anti-HNF4A (Rabbit monoclonal) | Cell Signaling Technology | Cat# 3113 | (1:400) |
| Antibody | Anti-MCM2 (Rabbit monoclonal) | Abcam | Cat# ab108935 | (1:5000) |
| Antibody | Anti-PDX1 (Mouse monoclonal) | DSHB | Cat# F109-D12 | (1:20) |
| Antibody | Anti-RFP (Rabbit polyclonal) | Rockland Immunochemicals | Cat# 600-401-379 | (1:1200) |
| Antibody | Anti-β-catenin (Mouse monoclonal) | BD Biosciences | Cat# 610153 | (1:200) |
| Antibody | Anti-Non-phospho β-catenin (Rabbit monoclonal) | Cell Signaling Technology | Cat# 8814 | (1:1600) |
| Antibody | Anti-Vinculin (Rabbit monoclonal) | Abcam | Cat# ab129002 | (1:20,000) |

*Continued*

| Reagent type (species) or resource | Designation | Source or reference | Identifiers | Additional information |
|---|---|---|---|---|
| Antibody | Anti-β-tubulin (Mouse monoclonal) | DSHB | Cat# E7 | (1:15,000) |
| Antibody | Anti-Axin1 (Rabbit monoclonal) | Cell Signaling Technology | Cat# 2087 | (1:1000) |
| Antibody | Anti-TCF4 (Rabbit monoclonal) | Cell Signaling Technology | Cat# 2569 | (1:1000) |
| Antibody | Anti-pFAK Y397 (Rabbit polyclonal) | Sigma Aldrich | Cat# SAB4504403 | (1:1000) |
| Antibody | Anti-Galectin 4 (Goat polyclonal) | R and D Systems | Cat# AF2128 | (1:200) |
| Antibody | Anti-Cathepsin E (Rabbit polyclonal) | Lifespan Biosciences | Cat# LSB523 | (1:12,000) |
| Antibody | Anti-BrdU (Rat monoclonal) | Abcam | Cat# ab6326 | (1:400) |
| Antibody | Anti-CyclinD1 (Rabbit monoclonal) | Abcam | Cat# ab137875 | (1:100) |
| Antibody | Anti-CyclinD2 (Mouse monoclonal) | NeoMarkers | Cat# MS-213-P1ABX | (1:500) |
| Other | PGK-Cre | *DuPage et al., 2009* PMID:19561589 | | Lentivirus |
| Other | Ad5CMVFlpo | Gene Transfer Vector Core, University of Iowa, IA | VVC-U of Iowa-530HT | Adenovirus |
| Other | Ad5CMVCre | Gene Transfer Vector Core, University of Iowa, IA | VVC-U of Iowa-5-HT | Adenovirus |
| Other | Ad5SpcCre | Gene Transfer Vector Core, University of Iowa, IA | VVC-Berns-1168 | Adenovirus |
| Other | *Dusp6* | ThermoFisher Scientific | Mm00518185_m1 | TaqMan Gene Expression Assay (FAM) |
| Other | *Pgc* | ThermoFisher Scientific | Mm00482488_m1 | TaqMan Gene Expression Assay (FAM) |
| Other | *Cym* | ThermoFisher Scientific | Mm01204823_m1 | TaqMan Gene Expression Assay (FAM) |
| Other | *Axin2* | ThermoFisher Scientific | Mm00443610_m1 | TaqMan Gene Expression Assay (FAM) |
| Other | *Lgr5* | ThermoFisher Scientific | Mm00438890_m1 | TaqMan Gene Expression Assay (FAM) |
| Other | *Gif* | ThermoFisher Scientific | Mm00433596_m1 | TaqMan Gene Expression Assay (FAM) |
| Other | *Ppia* | ThermoFisher Scientific | Mm02342429_g1 | TaqMan Gene Expression Assay (FAM) |
| Other | *Gfi1b* | ThermoFisher Scientific | Mm00492318_m1 | TaqMan Gene Expression Assay (FAM) |
| Other | *Pou2f3* | ThermoFisher Scientific | Mm00478293_m1 | TaqMan Gene Expression Assay (FAM) |
| Other | *Ccnd1* | ThermoFisher Scientific | Mm00432359_m1 | TaqMan Gene Expression Assay (FAM) |
| Other | *Ccnd2* | ThermoFisher Scientific | Mm00438070_m1 | TaqMan Gene Expression Assay (FAM) |

*Continued on next page*

*Continued*

| Reagent type (species) or resource | Designation | Source or reference | Identifiers | Additional information |
|---|---|---|---|---|
| Other | *Foxa1* | ThermoFisher Scientific | Mm00484713_m1 | TaqMan Gene Expression Assay (FAM) |
| Other | *Foxa2* | ThermoFisher Scientific | Mm01976556_s1 | TaqMan Gene Expression Assay (FAM) |
| Other | RBC Lysis Buffer | eBioscience | 00-4333-57 | |
| Other | Collagenase type I | Thermofisher Scientific | 17100017 | Enzyme |
| Other | Dispase | Corning | 354235 | Enzyme |
| Other | Deoxyribonuclease I | Sigma Aldrich | DN25 | Enzyme |

## Mice, tumor initiation, and drug treatment in vivo

All animal work was done in accordance with a protocol approved by the University of Utah Institutional Animal Care and Use Committee. The following mouse strains were used. $Braf^{LSL-V600E/+}$ (*Dankort et al., 2007*), $Braf^{FSF-V600E/+}$ (*Shai et al., 2015*), $Trp53^{f/f}$ (*Jonkers et al., 2001*), $Trp53^{frt/frt}$ (*Lee et al., 2012*), $Kras^{LSL-G12D/+}$ (*Jackson et al., 2001*), $Kras^{FSF-G12D/+}$ (*Young et al., 2011*), $Nkx2-1^{f/f}$ (*Kusakabe et al., 2006*), $Rosa26^{LSL-tdTomato}$ (*Madisen et al., 2010*), $Rosa26^{FSF-CreERT2}$ (*Schönhuber et al., 2014*), $Foxa1^{f/f}$ (*Gao et al., 2008*), $Foxa2^{f/f}$ (*Sund et al., 2000*). All animals were maintained on a mixed C57BL/6J × 129SvJ background. Tumors were generated by administering mice with PGK-Cre lentivirus at $5 \times 10^3$ plaque forming units (pfu)/mouse, Ad5CMVFlpO at $2 \times 10^7$ pfu/mouse; Ad5CMVCre $2.5 \times 10^7$ pfu/mouse, Ad5SpcCre 5–8 $\times 10^8$ pfu/mouse. Adenovirus was obtained from University of Iowa Viral Vector Core.

Rodent Lab Diet (AIN-76A) was formulated with the vemurafenib-related compound PLX4720 at 200 mg/kg (*Tsai et al., 2008*) and PD0325901 7 mg/kg (*Trejo et al., 2012*). Drug formulation was by Plexxikon and chow manufacture was by Research Diets. AIN-76A was used as control chow. Mice were maintained on the indicated drug pellets for the indicated time periods. BrdU incorporation was performed by injecting mice at 50 mg/kg (Sigma) intraperitoneally 1 hr prior to necropsy. Mice in survival studies were monitored for lethargy and respiratory distress, at which time animals were euthanized.

LGK-974 and Palbociclib (Selleck Chemicals) were delivered to mice via oral gavage. LGK-974 was formulated in 0.5% (w/v) methylcellulose/0.5% (v/v) Tween-80 solution and given at 7.5 mg/kg dose with 5 days ON/2 days OFF schedule for 2 weeks (*Figure 8C*, S8D) or 4 weeks (*Figure 8D and E*, S8E). Weights were monitored once or twice weekly and no toxicity was observed in mice. Palbociclib was formulated in 50 mM Sodium Lactate and initially given at 120 mg/kg dose with 5 days ON/2 days OFF schedule. Weights were monitored once or twice weekly. Due to weight loss in female mice receiving Palbociclib + MAPK-inhibitor chow combination, the following modifications were made for all mice receiving Palbociclib as single or dual agent. On week 2: 120 mg/kg dose with 4 days ON/3 days OFF schedule; week 3: Palbociclib was skipped altogether; week 4: 100 mg/kg dose with 4 days ON/3 days OFF schedule.

## Tamoxifen administration

Tumor-specific activation of $CreER^{T2}$ nuclear activity was achieved by intraperitoneal injection of tamoxifen (Sigma) dissolved in corn oil at a dose of 120 mg/kg. Mice received a total of six injections over the course of 9 days. For survival experiments, mice were additionally given pellets supplemented with 500 mg/kg tamoxifen (Envigo).

## Cell lines and cell culture

HEK-293T cells were cultured in DMEM/High Glucose medium (Gibco). L-WRN cells (ATCC) were cultured in DMEM/High Glucose medium containing G418 (Sigma) and Hygromycin (InvivoGen) media. All media contained 10% FBS (Sigma). To eliminate mycoplasma contamination, cell lines were treated with 25 µg/mL Plasmocin (InvivoGen) for 2–3 weeks. To maintain cultures mycoplasma free, media were supplemented with 2.5 µg/mL Plasmocin. Cell line identity was authenticated using STR analysis at the University of Utah DNA Sequencing Core.

## Establishing primary lung organoids

Eight to 10 weeks after tumor initiation with PGK-Cre, tumor bearing mice were euthanized and lungs were isolated. Lungs were then minced under sterile conditions and digested at 37°C for 30 mins with continuous agitation in a solution containing the enzymes, Collagen Type I (Thermo Fisher Scientific, at 450 U/ml final); Dispase (Corning, at 5 U/ml); DNaseI (Sigma, at 0.25 mg/ml) and Antibiotic-Antimycotic solution (Gibco). The digested tissue was passed through an 18 or 20-gauge syringe needle. Enzyme reactions were then stopped by addition of cold DMEM/F-12 with 10% FBS and the cell suspension was dispersed through 100 μm, 70 μm, and 40 μm series of cell strainers to obtain single-cell suspension. Subsequently, cell pellets were treated with erythrocyte lysis buffer (eBioscience). Finally, cell pellets were reconstituted in Advanced DMEM/F-12 (Gibco) supplemented with L-glutamine, 10 mM HEPES, and Antibiotic-Antimycotic. Thereafter, 100,000 tumor cells were mixed with 50 μl of Matrigel (Corning) at 1 to 10 ratio and plated in 24-well plates. For the first week of organoid initiation, Matrigel droplets were overlaid with recombinant organoid medium (Advanced DMEM/F-12 supplemented with 1X B27 (Gibco), 1X N2 (Gibco), 1.25 mM nAcetylcysteine (Sigma), 10 mM Nicotinamide (Sigma), 10 nM Gastrin (Sigma) and containing growth factors (100 ng/ml EGF [Peprotech], 100 ng/ml R-spondin1 [Peprotech], 100 ng/ml Noggin [Peprotech], 100 ng/ml FGF10 [Peprotech], as well as the ROCK inhibitor Y27632 [R and D Systems] and the TGF-β type I receptor-inhibitor SB431542 [R and D Systems]) as described in *Barker et al., 2010*; *Sato et al., 2009*).

After the initial spheroid culture propagation, organoids were switched to 50% L-WRN conditioned media generated as detailed in *Miyoshi and Stappenbeck, 2013*. Briefly, L-WRN cells (ATCC) seeded and maintained in 10% FBS, DMEM high glucose containing 500 μg/ml G418 (Sigma) and 500 μg/ml Hygromycin (InvivoGen) media until confluency. Once confluent, L-WRN cells were switched to Advanced DMEM/F-12 containing 20% FBS, 2 mM L-glutamine, 100 U/ml penicillin, 0.1 mg/ml streptomycin. Daily, conditioned supernatant medium was collected from L-WRN cultures, filtered through 0.2 μm vacuum filters and saved at −20°C as stock (100%) L-WRN medium. To use for culturing spheroids, stock L-WRN was diluted with equal volume Advanced DMEM/F-12 (final concentration 50%). Where specified, stock L-WRN media was diluted with Advanced DMEM/F-12 to a final concentration of 5% while still keeping FBS concentration at 10% as described in *VanDussen et al., 2015*. For drug studies, organoid media was supplemented with Cobimetinib (GDC-0973), GDC-0994 (Genentech), LGK-974, PD0325901, PLX-4720 (Selleck Chemicals) at indicated concentrations. Drug containing media was refreshed every 48–72 hr.

## Lentiviral production and transduction

HEK293T cells were transfected with CRE-encoding lentiviral vector, d8.9 packaging vector and VSV-G envelope vector mixed with TransIT-293 (Mirus Bio). Virus containing supernatant was collected at 36, 48, 60, and 72 hr after transfection. Ultracentrifugation at 25,000 r.p.m. for 2 hr was necessary to concentrate virus for in vivo infection (*DuPage et al., 2009*).

For stable transduction of organoids, organoid cultures were first prepared into single cell suspensions by subjecting them to successive incubations with Cell Recovery Solution (Corning) and TrypLE (Gibco). Cells were then resuspended in a 1:1 mixture, by volume, of 50% L-WRN and lentivirus containing supernatant. After supplementation with 8 μg/ml polybrene, cells were incubated for 2 hr. Cells were then pelleted, mixed back with Matrigel, and seeded. 72 hr later, Puromycin selection for 1 week was performed to achieve stable lines.

## Flank tumor transplantation

For subcutaneous allograft experiments, $3 \times 10^5$ single-cell suspension of 988 BPN organoid cells stably expressing either the Mission pLKO.1-puro Non Target shRNA control (shscr) or one of the validated *Ccnd2* Mission shRNAs, TRCN0000054764 (#64) or TRCN0000054766 (#66), were mixed with in 1:1 vol with 50 μl Matrigel. Cells were injected subcutaneously into the right or left flanks of NOD/SCID-gamma chain deficient (NSG) mice. Tumor dimensions were measured with calipers. Tumor volume was calculated using the (L x $W^2$)/2 formula. When the majority of the tumors reached ~200 mm³, mice were randomized into control or drug groups, the latter of which received RAF/MEK inhibitor compounded chow for 4 weeks. Tumor volume measurements were not reliable as the tumors formed from organoid cultures became fluid-filled cysts rather than solid. At end of treatment period, tumors were removed and processed for histological analyses.

## Real-time PCR

Total RNA was isolated using TRIzol followed by PureLink RNA Mini Kit (Thermo Fisher Scientific). RNA was treated with RNase-Free DNAse I (Invitrogen) on-column. For low cell numbers (e.g. after FACS), Animal Tissue RNA Purification Kit (Norgen Biotek) was used instead. RNA was converted to cDNA using iScript Reverse Transcription Supermix (BioRad). cDNA was analyzed either by SYBR green real-time PCR with SsoAdvanced Universal SYBR Green Supermix (BioRad) or by Taqman real-time PCR with SsoAdvanced Universal Probes Supermix (BioRad) using a CFX384 Touch Real-Time PCR Detection System (BioRad). Gene expression was calculated relative to *Ppia* (loading control) using the $2^{-(dCt.x-average(dCt.control))}$ method and normalized to the control group for graphing. Primer sequences and TaqMan Assays used in this study are listed below.

> *Ppia* Mm02342429_g1; *Pgc* Mm00482488_m1; *Cym* Mm01204823_m1; *Gif* Mm00433596_m1
> *Pou2f3* Mm00478293_m1; *Gfi1b* Mm00492318_m1; *Lgr5* Mm00438890_m1
> *Axin2* Mm00443610_m1; *Ccnd1* Mm00432359_m1; *Ccnd2* Mm00438070_m1
> *Foxa1* Mm00484713_m1; *Foxa2* Mm01976556_s1; *Dusp6* Mm00518185_m1

## Cell viability and growth assay

For organoids, cells were seeded at a density of 2000 cells per 15 µl Matrigel dome, four domes per line per time point, each dome in a single well of a 96-well plate. After overnight culture, organoids were treated with different concentrations of the indicated inhibitors and incubated at 37°C for indicated times. Subsequently, the relative number of viable cells was measured by by CellTiter-Glo 3D Cell Viability Assay (Promega), according to the manufacturer's instructions. Luminescence was then measured by a microplate reader (EnVision 2105 Multimode Plate Reader, PerkinElmer). Replicate values for each experimental group were averaged and all values were normalized to control treatment group for each line.

## Immunoblot analysis

Cells were lysed on ice using RIPA buffer (50 mM HEPES, 140 mM NaCl, 1 mM EDTA, 1% triton X-100, 0.1% sodium deoxycholate and 0.1% SDS) supplemented with protease and phosphatase inhibitor cocktails (Roche). Protein extracts were clarified and concentrations were measured with Pierce Coomassie Plus Protein Assay reagent (Thermo Fisher Scientific). Lysates were then resolved on SDS-PAGE gels (BioRad), and transferred to Nitrocellulose blots. Membranes were probed with primary antibodies against AXIN1 (2087, CST), TCF4 (2569, CST) pFAK Y397 (SAB4504403, Sigma), DUSP6 (EPR129Y, Abcam), SPRY2 (EPR4318(2)(B), Abcam), pERK (4370, CST), tERK (4696, CST), Vinculin (EPR8185, Abcam), and β-tubulin (E7, DSHB). Blots were subsequently incubated with HRP conjugated secondary antibodies. For signal development, Supersignal West Femto Substrate kit (Thermo Fisher Scientific) was used, followed by image acquisition using darkroom development. ImageJ was used for band intensity quantitation.

## Histology and immunohistochemistry

Lung tissues were fixed overnight in 10% neutral buffered formalin, processed through 70% ethanol, embedded in paraffin, and sectioned at 4 µm thickness. Staining of hematoxylin and eosin, as well as detection of mucin by Alcian Blue were carried out at the HCI Research Histology Shared Resource. Immunohistochemistry was performed manually as detailed in *Camolotto et al., 2018*. The following primary antibodies were used. BrdU (BU1/75, Abcam), Cathepsin E (LS-B523, Lifespan Biosciences), Galectin 4 (AF2128, R and D Systems), Gastrokine 1 (2E5, Abnova), Histone H3 pSer10 (ab5176, Abcam), HNF4A (C11F12, CST), MCM2 (ab31159, Abcam), Muc5AC (SPM488, Abnova), NKX2-1 (EP1584Y, Abcam), PDX1 (F109-D12, DSHB), RFP (600-401-379, Rockland Immunochemicals), pERK1/2 (D13.14.4E, CST), pRSK S380 (D3H11, CST), p4EBP1 T37/46 (2855, CST), pS6 S235/236 (D57.2.2E, CST), pS6 S240/244 (D68F8, CST), pMEK1/2 S221 (166F8, CST), Pepsinogen C (HPA031718, Sigma), POU2F3 (HPA019652, Sigma), pRB S807/811 (8516, CST), Cyclin D1 (SP4, Abcam), Cyclin D2 (DCS-3.1, NeoMarkers), CD36 (324205, R and D Systems), pro-SPB (AB3430, Millipore), pro-SPC (AB3786, Millipore), β-catenin (610153, BD Biosciences), active β-catenin (D13A1, CST). To visualize Cyclin D2 signal, IHC was performed using the CSA II, Biotin-Free Catalyzed Amplification System (Dako) following manufacturer's instructions. Pictures were taken on a

Nikon Eclipse Ni-U microscope with a DS-Ri2 camera and NIS-Elements software. Tumor quantitation was performed on hematoxylin and eosin-stained or IHC-stained slides using NIS-Elements software. All histopathologic analyses were performed by a board-certified anatomic pathologist (E.L. S.).

## Primary human tumors

De-identified formalin fixed, paraffin-embedded (FFPE) tumors were obtained from the Intermountain Biorepository, which collects samples in accordance with protocols approved by the Intermountain Healthcare Institutional Review Board. Mucinous tumors were included in the study based on the following criteria: 1. Diagnosis of invasive mucinous adenocarcinoma; OR 2. Diagnosis of mucinous bronchioloalveolar carcinoma (older diagnostic term for IMA); OR 3. Mucinous lung adenocarcinoma with strong morphologic features of IMA, such as abundant intracellular mucin. Colloid adenocarcinomas were excluded, as these are considered to be a discrete subtype of lung cancer.

## Fluorescence-activated cell sorting (FACS) for in vivo tumors and in vitro organoid cultures

Tumor bearing lungs were isolated in ice-cold PBS. Lungs were enzymatically digested as described above in the procedure for establishing primary organoids. Once single-cell suspensions were obtained, cells were reconstituted in phenol red-free DMEM/F-12 with HEPES containing 2% FBS, 2% BSA and DAPI. Cells were sorted using BD FACS Aria for tdTomato-positive and DAPI-negative cells into DMEM/F-12% and 30% serum. After sorting, cells were pelleted and flash frozen or resuspended TRIzol then frozen at −80°C.

To study the dynamics of WNT signaling in vitro, BPN organoids were stably transduced with the β-catenin/TCF-dependent GFP reporter lentiviral construct 7TGP (Addgene plasmid #24305). Single cells were isolated from Matrigel using Cell Recovery Solution and TrypLE incubations. Cells were analyzed on BD LSRFortessa or BD FACSAria and sorted on BD FACSAria. Events first passed through a routine light-scatter and doublet discrimination gates, followed by exclusion of DAPI-positive dead cells. Viable tumor organoids were further identified as tdTomato-high cells. 7TGP organoids grown in 5% L-WRN media for 2–3 days were used as control for setting the GFP-negative gate. This gating strategy was applied to 7TGP organoids that were cultured in standard 50% L-WRN media in order to identify and sort for the WNT-reporter-low fraction. This fraction was collected in 50% L-WRN and expanded for ~1 week. After expansion, sorted 7TGP organoids were seeded and treated with 50% L-WRN (± Cobimetinib) or 5% L-WRN (± Cobimetinib) for 24 hr, 48 hr, and 72 hr. Then, the fraction of WNT-reporter-high population under the above four conditions was analyzed. Flow cytometric data analyses was performed using FlowJo software.

## RNA sequencing from whole tumors

Library preparation was performed using the TruSeq Stranded mRNA Library Preparation Kit with poly(A) selection (Illumina). Purified libraries were qualified on an Agilent Technologies 2200 TapeStation using a D1000 ScreenTape assay. The molarity of adapter-modified molecules was defined by quantitative PCR using the Kapa Biosystems Kapa Library Quant Kit. Individual libraries were normalized to 5 nM and equal volumes were pooled in preparation for Illumina sequence analysis. Libraries were sequenced on Illumina HiSeq 2500 (50 cycle single-read sequencing v4).

## Whole-tumor RNA-seq analysis

Adapters in raw FASTQ files containing 50 bp single-end reads were trimmed with cutadapt. QC metrics were generated for each sample with FastQC and Picard's CollectRnaSeqMetrics after aligning reads to genome with STAR (*Dobin et al., 2013*). QC metrics were summarized by MultiQC. From this QC analysis, one sample (id 14489 × 13) did not group with other samples in the MultiQC report. It was later confirmed that very limited starting material was a contributing factor to its difference from the others. This sample was removed from downstream analysis because the noise in the sample was likely a larger negative trade-off than the gains from increasing biological replicates for that condition.

Next, Salmon v0.9.1 quantified RNA transcript expression with two steps (*Patro et al., 2017*); first, the mouse transcriptome downloaded from Gencode (release 26) was indexed using k-mers of

length 19 with type 'quasi'; second, Salmon quasi-mapped reads and corrected for sequence-specific bias (see option –seqBias). Salmon-based transcript expression estimates were converted to gene expression estimates with R package tximport (*Soneson et al., 2015*).

Differential gene expression modeling with DESeq2 (*Love et al., 2014*) and EdgeR (*Robinson et al., 2010*) evaluated four distinct contrasts testing (i) genotype effects within control groups [BP C vs. BPN C]; (ii) genotype effects within treatment groups [BP Tx vs. BPN Tx]; (iii) treatment effects within BP genotype [BP C vs. BP Tx]; (iv) treatment effects within BPN genotype [BPN C vs. BPN Tx]. Prior to fitting a single-factor DESeq2 regression (four factor levels: BP C, BPN C, BP Tx, BPN Tx), genes were filtered if there were less than five total counts across all samples. PCA was done on the top 500 most variable genes subject to a regularized log transform to confirm within-group variation was similar. A Wald test was applied to the fitted model for each of the four contrasts specified above, where the null hypothesis was gene expression differences less than or equal to log2(2) fold change in absolute value and the alternative hypothesis was that differences exceed log2(2). Significance of gene-wise differences was controlled by a false discovery rate of 10% (*Benjamini and Hochberg, 1995*).

Differential gene expression modeling with EdgeR for volcano plots used package edgeR_3.24.3 with ggplot2_3.1.1. Significance of gene-wise differences was calculated by tag-wise exact test. WNT signaling genes were delineated in the whole-tumor sequencing matrix using AmiGO pathway annotations (http://amigo.geneontology.org) for 'Wnt Signaling' (*Carbon et al., 2009*). WNT genes were restricted to those annotated for *Mus musculus* with direct experimental evidence. The most differentially expressed genes between treated and untreated sample groups (Bonferonni corrected Student's t-test) were thresholded at >2 fold change between means of treated and untreated samples. The 13 genes meeting this criterion and showing upregulation in RAF/MEK inhibitor-treated cells (BPN Tx) was designated as the WNT13 signature in this study.

Gene set enrichment analysis (GSEA) was carried out on RNA-seq data from whole tumors (and single cell data (below)) by comparing gene expression profiles with archived gene sets from Hallmarks [ftp.broadinstitute.org://pub/gsea/gene_sets/h.all.v7.0.symbols.gmt] and Oncogenic signatures [ftp.broadinstitute.org://pub/gsea/gene_sets/c6.all.v7.0.symbols.gmt] as well as with the cell type specific genes sets described in *Haber et al., 2017*; *Leushacke et al., 2017*; *Montoro et al., 2018*; *Zhang et al., 2019*.

## Single-cell RNA sequencing

All protocols to generate scRNA-seq data on 10x Genomics Chromium platform including library prep, instrument and sequencing setting can be found at: https://support.10xgenomics.com/single-cell-gene-expression.

The Chromium Single Cell Gene Expression Solution with 3' chemistry, version 3 (PN-1000075) was used to barcode individual cells with 16 bp 10x Barcode and to tag cell specific transcript molecules with 10 bp Unique Molecular Identifier (UMI) according to the manufacturer's instructions. The following protocol was performed at High-Throughput Genomics Shared Resources at Huntsman Cancer Institute, University of Utah. First, FACS sorted single-cell suspensions of tdTomato-positive lung tumors were resuspended in phosphate buffered saline with 0.04% bovine serum albumin. The cell suspension was filtered through 40 micron cell strainer. Viability and cell count were assessed on Countess I (Thermo Scientific). Equilibrium to targeted cell recovery of 6,000 cells, along with 10x Gel Beads and reverse transcription reagents were loaded to Chromium Single Cell A (PN-120236) to form Gel Beads-in emulsions (GEMs), the nano-droplets. Within individual GEMs, cDNA generated from captured and barcoded mRNA was synthesized by reverse transcription at the setting of 53°C for 45 min followed by 85°C for 5 min. Subsequent A tailing, end repair, adaptor ligation and sample indexing were performed in bulk according to the manufacturer's instructions. he resulting barcoding libraries were qualified on Agilent D1000 ScreenTape on Agilent Technology 2200 TapeStation system and quantified by quantification PCR using KAPA Biosystems Library Quantification Kit for Illumine Platforms (KK4842). Multiple libraries were then normalized and sequenced on NovaSeq 6000 with 2 × 150 PE mode.

## Single-cell RNA-seq analysis

### Demultiplexing and data alignment

Single-cell RNA-seq data were demultiplexed with 10x cellranger mkfastq version 3.1.0 to create fastq files with the I1 sample index, R1 cell barcode+UMI and R2 sequence. Reads were aligned to the mouse genome (mm10 with custom tdTomato reference) and feature counts were generated using cellranger count 3.1.0 with expected-cells set to 6000 per library. QC reporting, clustering, dimension reduction, and differential gene expression analysis were performed in 10x Genomics' Cell Loupe Browser (v3.1.1). For the BPN control sample, we captured 5065 cells and obtained 46,894 mean reads per cell; 3544 median genes per cell. For the BPN treated sample, we captured 5563 cells and obtained 37,488 mean reads per cell; 2771 median genes per cell. For further details of the primary Cell Ranger data processing, see https://support.10xgenomics.com/single-cell-gene-expression/software/pipelines/latest/algorithms/.

### Visualization and expression analysis

Single-cell expression data was analyzed using Seurat (3.1.0) (*Butler et al., 2018*; *Stuart et al., 2019*) and visualized in Loupe browser (3.1.1). Alternatively, single-cell data was processed mostly in R (3.5.1). Cells with unique feature counts over 7500 or less than 200 and more than 20% mitochondrial counts have been filtered out. For visualization, the umap-learn package, from Seurat (*McInnes et al., 2018*), was used to reduce the dimensionality of the data.

### Clustering

The Seurat package clustering method was used to designate cluster membership of single cell transcriptomics profiles. Seurat clustering is heavily inspired by *Levine et al., 2015* and *Xu and Su, 2015*, which has two steps in the first step, the findNeighbors function is employed to embed single-cell profiles in a K-nearest neighbor (KNN) graph based euclidean distance in a (20 PCs) PCA space, followed by refining the edge weights based on jaccard similarity (https://satijalab.org/seurat/v3.0/pbmc3k_tutorial.html). In the second step, the FindClusters function employs Louvain algorithm modularity optimization techniques to iteratively group cells together (https://satijalab.org/seurat/v3.0/pbmc3k_tutorial.html).

### Stromal cell designations

Clusters of cells sharing highly similar positions in dimensionality reduced UMAP-space were examined for differential expression using Loupe browser's built-in differential expression calculator, and genes showing significant cluster specific expression were screened manually against the literature for markers of known cell types. Those clusters representing minority populations that were highly distinct in Umap-space from the bulk of cells in the analysis and which bear distinctive expression of stromal marker genes were considered as stromal contaminants and removed from subsequent, tumor-specific Umap data visualization and cluster specific gene expression level analysis.

### Differential gene expression using seurat

For the DEG test the 'FindMarkers' function from the Seurat package with default setting was used, including Wilcoxon Rank Sum test, and Bonferroni multi-testing correction based on the total number of genes in the dataset.

### Cell cycle phase modeling

We calculated cell cycle scores of control and treated tumor cells in scRNA-seq data using the Seurat package and the cell cycle genes defined by *Mizuno et al., 2009*. The Seurat package uses the scoring strategy described in *Tirosh et al., 2016* and assigns scores for cells based on the expression of G2/M and S phase marker genes, with low scoring cells defaulting to G1. In an alternative approach, we modeled cell cycle phase by determining mean expression per cell of each phased cell cycle signature described in *Mizuno et al., 2009*. A matrix of these values was then graphed as a diffusion map using the Destiny package (*Angerer et al., 2016*) in R. A coherent cell cycle loop through three dimensional space using the first three diffusion components was projected onto two dimensions using an heuristic arithmetic, diffusion component combination. To this graph a principal curve was

fit using Princurve (*Hastie and Stuetzle, 1989*) and tangents for each point (lambdas) were used as cellular positions along a trajectory designated as Q-depth based on the observed position of G0-signature enriched cells (an indicator related to quiescence) at the extreme distal to cells enriched in G1 signatures or the S and G2/M signatures.

### MEK activation signature and WNT signature expression

Average expression of an 18 gene signature from *Dry et al., 2010* was determined per cell across the single cell data set to model MEK activation state. Enrichment for the WNT13 signature described above was similarly examined across the single-cell dataset.

## Data imputation

We used a novel data imputation approach to model the expression of low-detection genes in single-cell data (https://github.com/TheSpikeLab/FIESTA; *Mehrabad et al., 2021a*; *Mehrabad et al., 2021b*). Briefly, a weighted non-negative matrix factorization was carried out on the single-cell data matrix using machine-learning-based factor optimization. Subsequently, optimized factors were used to reconstruct an idealized completed matrix with values recommended for many gene nodes that were originally below the detection threshold.

## Recombination fidelity

Cluster-specific BAM files were parsed from the concatenated scRNA-Seq BAM files using subset-bam tool (https://github.com/10XGenomics/subset-bam, *10xGenomics, 2020*). The presence of select splice junctions was visualized using IGV (version 2.7.2) (*Robinson et al., 2011*) against mouse genome version (mm10).

Statistics p-values were calculated using the unpaired two-tailed t-test. RNA-seq statistics are described above.

## Acknowledgements

We are grateful to members of the Snyder lab for suggestions and comments. We acknowledge Plexxikon for providing PLX4720 and PD0325901; Research Diets for compounding. We thank Brian Dalley and Opal Allen for sequencing expertise, Chris Conley for RNA sequencing data analysis expertise, James Marvin for FACS expertise and David H Lum and members of the Preclinical Research Shared Resource core for in vivo drug administration. Work in the flow cytometry core was supported by the National Center for Research Resources of the National Institutes of Health under Award Number 1S20RR026802-1. Research reported in this publication utilized shared resources (including Flow Cytometry, High Throughput Genomics, Bioinformatics, Biorepository and Molecular Pathology, and Preclinical Research Resource) at the University of Utah and was supported by Huntsman Cancer Foundation and the National Cancer Institute of the National Institutes of Health under Award Number P30CA042014. ELS was supported in part by a Career Award for Medical Scientists from the Burroughs Wellcome Fund, a V Scholar Award, the NIH (R01CA212415, R01CA240317), a Pilot Project Award from the Lung Cancer Center of the Huntsman Cancer Institute, and institutional funds (Department of Pathology and Huntsman Cancer Institute, University of Utah). GO was supported by the NIH (F31CA243427 and the Eunice Kennedy Shriver National Institute of Child Health and Human Development of the National Institutes of Health under Award Number T32HD007491). The content is solely the responsibility of the authors and does not necessarily represent the official views of the NIH.

## Additional information

### Funding

| Funder | Grant reference number | Author |
|---|---|---|
| V Foundation for Cancer Research | V Scholar Award | Eric L Snyder |
| Burroughs Wellcome Fund | Career Award for Medical | Eric L Snyder |

| | | Scientists |
|---|---|---|
| National Cancer Institute | R01CA212415 | Eric L Snyder |
| National Cancer Institute | R01CA240317 | Eric L Snyder |
| National Cancer Institute | F31CA243427 | Grace Orstad |
| Huntsman Cancer Institute | Pilot Project Award | Benjamin T Spike<br>Eric L Snyder |
| Eunice Kennedy Shriver National Institute of Child Health and Human Development | T32HD007491 | Grace Orstad |

The funders had no role in study design, data collection and interpretation, or the decision to submit the work for publication.

## Author contributions

Rediet Zewdu, Conceptualization, Formal analysis, Investigation, Visualization, Writing - original draft, Writing - review and editing, Performed almost all experiments; Elnaz Mirzaei Mehrabad, Formal analysis, Visualization, Writing - review and editing, Performed all computational analysis for RNA sequencing; Kelley Ingram, Pengshu Fang, Katherine L Gillis, Soledad A Camolotto, Grace Orstad, Alex Jones, Investigation; Michelle C Mendoza, Supervision, Funding acquisition, Writing - review and editing; Benjamin T Spike, Formal analysis, Supervision, Funding acquisition, Methodology, Writing - original draft, Writing - review and editing, Performed all computational analysis for RNA sequencing; Eric L Snyder, Conceptualization, Resources, Formal analysis, Supervision, Funding acquisition, Investigation, Writing - original draft, Project administration, Writing - review and editing

## Author ORCIDs

Rediet Zewdu (iD) https://orcid.org/0000-0001-6784-6068
Michelle C Mendoza (iD) http://orcid.org/0000-0002-6490-1794
Eric L Snyder (iD) https://orcid.org/0000-0003-3591-3195

## Ethics

Animal experimentation: All animal work was done in accordance with a protocol (#18-08005) approved by the University of Utah Institutional Animal Care and Use Committee.

## Decision letter and Author response

Decision letter https://doi.org/10.7554/eLife.66788.sa1
Author response https://doi.org/10.7554/eLife.66788.sa2

# Additional files

## Supplementary files

- Supplementary file 1. Whole-tumor RNA sequencing DESeq2 analysis.
- Supplementary file 2. Differentially expressed genes between control and MAPKi-treated tumors used for generating volcano plots.
- Supplementary file 3. Top 10 upregulated genes per tumor cluster in scRNA-seq data.
- Supplementary file 4. Full list of DEGs from single cell sequencing analysis.
- Supplementary file 5. GO analysis by Correlation Engine comparing BP C and BP Tx tumors.
- Supplementary file 6. GO analysis by Correlation Engine comparing BPN C and BPN Tx tumors.
- Supplementary file 7. Published gene expression dataset for gastric chief cells.
- Supplementary file 8. Published gene expression dataset for tuft cells.
- Transparent reporting form

## Data availability

All sequencing data generated in this study are available at Gene Expression Omnibus (GEO: GSE145152).

The following dataset was generated:

| Author(s) | Year | Dataset title | Dataset URL | Database and Identifier |
|---|---|---|---|---|
| Zewdu R, Mehrabad EM, Ingram K, Fang P, Gillis KL, Camolotto SA, Orstad G, Jones A, Mendoza MC, Spike BT, Eric L, Snyder EL | 2021 | An NKX2-1/ERK/WNT feedback loop modulates gastric identity and response to targeted therapy in lung adenocarcinoma | https://www.ncbi.nlm.nih.gov/geo/query/acc.cgi?acc=GSE145152 | NCBI Gene Expression Omnibus, GSE145152 |

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
