## [Decision Letter]

**Acceptance summary:**

This manuscript from the Snyder laboratory details cell lineage states that are controlled by NKX2-1 and oncogenic MAPK signaling in BRAFV600E-driven lung cancers. The work builds on previous works from the Snyder group showing NKX2-1 suppresses a latent gastric differentiation program in KRASG12D-driven lung cancers. Switching the model from KRAS to BRAF, now the Snyder laboratory demonstrates multiple similarities between the oncogenic drivers and details key differences that have significant impact on our understanding of lung cancer etiology and possibly treatment. The depth of data analysis and breadth of methodology used represent a real tour de force in cancer modeling. The insights highlight the complex interplay between mitogenic signaling and developmentally-related pathways during cancer progression. The insights gleaned from the study have some potential in influence treatment strategies. As such, this study will appeal to a broad audience and has sufficient impact for publication in the *eLife* journal. The stated conclusions from the work are entirely sound and wholly supported by the data presented.

**Decision letter after peer review:**

Thank you for submitting your article "An NKX2-1/ERK/WNT feedback loop modulates gastric identity and response to targeted therapy in lung adenocarcinoma" for consideration by *eLife*. Your article has been reviewed by 3 peer reviewers, and the evaluation has been overseen by Maureen Murphy as the Senior and Reviewing Editor. The following individual involved in review of your submission has agreed to reveal their identity: Monte M Winslow (Reviewer #2).

Essential Revisions:

1. It seems that NKX2-1 is controlling a sort of negative feedback pathway on MAPK signaling (Figure 3). What is the nature of the feedback? Does this perhaps account for the reduced initiation of BRAFV600E driven tumors; i.e. does simultaneous loss of NKX2-1 and activation of BRAFV600E initiate a senescence program that effectively blocks tumor initiation?

2. Does the cessation of MAPKi treatments in BPN tumors lead to a transition in cell state back to a gastric-like cell state or are the changes toward chief/tuft-like cell states durable? Insights beyond survival from samples in Supplemental Figure 8E might be helpful.

3. The authors should perform RNA in situ hybridization to identify the probable source of WNT in BPN mouse models treated with MAPK inhibitors.

4. Did the authors analyze the changes in immune cell infiltration in BP and BPN tumors after MAPK inhibition?

5. To understand the dependence of β-catenin in WNT-mediated effects, the authors need to analyze the activation of β-catenin in BP and BPN tumors treated with MAPK inhibitors. In addition, it would be interesting to see the effect of MAPKi+ LGK974 (Figure 8c) on phosphorylation of β-catenin.

6. The authors jump right into data on BP versus BPN tumors without providing any information on B versus BN tumors. If they have any data on those p53-proficient tumors it would be nice to add.

7. For the data in Figure 1B, if would be nice if the authors also reported tumors # and tumor size (area). They claim that there is a defect in tumor initiation, which I imagine means tumors number, but they neither show low magnification images or quantification.

---

## [Author Response]

Essential Revisions:1. It seems that NKX2-1 is controlling a sort of negative feedback pathway on MAPK signaling (Figure 3). What is the nature of the feedback? Does this perhaps account for the reduced initiation of BRAFV600E driven tumors; i.e. does simultaneous loss of NKX2-1 and activation of BRAFV600E initiate a senescence program that effectively blocks tumor initiation?

We agree with the reviewer that it is quite possible that concomitant BRAF^V600E^ activation and *Nkx2-1* deletion in normal AT2 cells leads to a high level of ERK activity that is not well tolerated by normal cells and either leads to senescence per se, or at least impairs tumor initiation to some degree. (In contrast, BRAF^V600E^-driven neoplastic cells that have already equilibrated to moderately increased ERK activity might more readily tolerate a further increase in ERK activity caused by *Nkx2-1* deletion.) Although we are not certain why this is true in the BRAF model but not the KRAS model, BRAF^V600E^ is thought to be a more potent activator of MAPK signaling than KRAS^G12D^ in certain contexts, and this may be one underlying reason for our observations. We have addressed this point in the revised Discussion.

In a previous study of the KRAS-driven model (Snyder et al. Mol Cell 2013), we have found that *Nkx2-1* deletion leads to a decline in two feedback inhibitors of MAPK signaling (*Dusp6* and *Spry2*). Both genes declined ~2 fold at the RNA level based on exon array data, and a decline in SPRY2 protein level was confirmed by IHC, but no functional studies of these feedback inhibitors were performed in that paper. In the current work, we find that BPN tumors also express lower levels of these genes than BP tumors. Changes in these genes are one potential explanation for the increased pERK observed in NKX2-1-negative mouse models. One of the authors (M. Mendoza) is leading a study to investigate the mechanisms by which NKX2-1 regulates these genes and the relevance of this feedback to lung adenocarcinoma. This study has been posted on Biorxiv (https://www.biorxiv.org/content/10.1101/2021.03.04.433941v1).

2. Does the cessation of MAPKi treatments in BPN tumors lead to a transition in cell state back to a gastric-like cell state or are the changes toward chief/tuft-like cell states durable? Insights beyond survival from samples in Supplemental Figure 8E might be helpful.

In the original manuscript (Figure 4—figure supplement 1), we show that BPN tumors transition back to their original morphology 2 weeks after drug withdrawal as assessed by morphology (H&E stain). In the revised manuscript, we provide additional analysis (Figure 6—figure supplement 2) demonstrating that GKN1 and mucin production (Alcian blue) return to the same levels as controls within 2 weeks after drug withdrawal. At this timepoint, PGC is undetectable and the percentage of POU2F3-positive cells is similar to controls. Thus, based on these markers of major cell states, BPN cells readily transition back to their original identity after drug withdrawal.

3. The authors should perform RNA in situ hybridization to identify the probable source of WNT in BPN mouse models treated with MAPK inhibitors.

To address this question, we have analyzed the levels of all murine Wnt genes in all cell clusters identified in scRNA-seq, including stromal cells (see Figure 4—figure supplement 1E). As presented in Figure 7—figure supplement 2, several *Wnt* genes were induced by MAPKi treatment in BPN tumor cells, including *Wnt2b*, *Wnt4*, *Wnt8b* and *Wnt10b*. Additional *Wnt* ligands such as *Wnt2*, *Wnt5b* and *Wnt6* were more highly expressed in stromal cells that were co-isolated during lung tumor preparation. Thus, there are multiple potential in vivo sources of WNT ligand, including both treated BPN tumor cells and the tumor stroma.

The observation that LGK-974 can block changes in cell identity induced by MEK inhibitor in BPN organoids in vitro (Figure 8) also suggests that at least some relevant WNT production occurs within BPN tumor cells.

4. Did the authors analyze the changes in immune cell infiltration in BP and BPN tumors after MAPK inhibition?

We did not formally analyze immune cell infiltration in these experiments. We believe that a comprehensive analysis (using a combination of microscopy, flow cytometry, CyTOF and/or scRNA-seq) would belong in a new manuscript. We anticipate that the direct impact of our drug regimen on MAPK signaling in immune cells would be relatively modest, given that PLX4720 has been reported to activate wild type RAS->RAF->MEK signaling, but PD0325901 should dampen MEK activation by this mechanism.

We have previously shown that *Nkx2-1* deletion in other lung cancer GEMMs leads to increased neutrophil infiltration, mediated in part by CXCL5 (DOI: 10.1016/j.immuni.2018.09.020). By H&E analysis, we observe accumulation of neutrophils in a subset of the glandular structures formed by control BPN tumors. (In contrast, neutrophils are much less frequent in BP tumors.) We can also readily identify macrophages and lymphocytes in control BPN tumors. These are generally located in the stroma between tumor glands, whereas neutrophils predominantly localize within glandular structures. Four weeks after treatment with BRAF/MEK inhibitors, it is very difficult to find neutrophils associated with residual BPN cells. Lymphocytes and macrophages can still be found in association with residual BPN cells, at least to some extent. We note that *Cxcl5* levels do not significantly change after treatment with BRAF/MEK inhibitors. One possible explanation for differences in immune infiltration could be that treated BPN cells no longer appear to be proliferating, invading and causing local tissue damage, which can induce an acute immune reaction. Treated BPN cells also no longer form glandular structures, whose three dimensional structure may be required for neutrophils to accumulate to high levels. We have elected not to elaborate on these findings in the revised manuscript as they do not affect our main conclusions, which are also supported by in vitro studies on tumor cells performed in the absence of an immune infiltrate. Rather, we believe these observations would be the starting point for a future in depth analysis of immune response in these models.

5. To understand the dependence of β-catenin in WNT-mediated effects, the authors need to analyze the activation of β-catenin in BP and BPN tumors treated with MAPK inhibitors. In addition, it would be interesting to see the effect of MAPKi+ LGK974 (Figure 8c) on phosphorylation of β-catenin.

Using antibodies for total and active (unphosphorylated) β-catenin (Figure 8—figure supplement 2), we find that control and LGK-974-treated BPN tumors exhibit predominantly membranous staining by IHC, with no evidence of nuclear β-catenin. MAPK inhibition alone leads to increased intensity throughout the cell, including the nucleus. Levels of active β-catenin in MAPKi+LGK974 treated tumors were lower than MAPKi-treated tumors. We did not detect a specific signal with an antibody for phospho-β-catenin (BS-3084R, data not shown).

Control BP tumors had lower levels of active β-catenin than BPN tumors, and there was no appreciable increase in staining with MAPK inhibition (Figure 8—figure supplement 2).

6. The authors jump right into data on BP versus BPN tumors without providing any information on B versus BN tumors. If they have any data on those p53-proficient tumors it would be nice to add.

Unfortunately, we do not have any data on p53 proficient (B vs. BN) tumors that we can include in the manuscript. We anticipate that the immediate impact of *Nkx2-1* deletion would be similar, but that (like K vs. KP tumors), the ability of these tumors to progress to a high grade state over time would be diminished.

7. For the data in Figure 1B, if would be nice if the authors also reported tumors # and tumor size (area). They claim that there is a defect in tumor initiation, which I imagine means tumors number, but they neither show low magnification images or quantification.

Unfortunately, the tumor burden is too high at this timepoint to accurately quantitate individual tumors. In many areas, multiple tumors have grown together and become confluent, thereby precluding accurate identification of individual tumors. We provide representative low power images corresponding to Figure 1B in the revised manuscript (Figure 1—figure supplement 1C). However, we consider the graph to be the most accurate representation of overall tumor burden given the heterogeneity in the model.

Based on this comment, we have changed our interpretation in the revised manuscript to state that *Nkx2-1* deletion at the time of initiation impairs the early stages of BRAFV600E-driven tumorigenesis. The impairment appears limited to early stages of tumorigenesis, quite likely the earliest events associated with transformation of normal lung epithelium, because BP and BPN tumors ultimately equilibrate to a similar proliferation rate, and BPN tumors can progress to higher grade adenocarcinomas that ultimately compromise lung function.